# FL-GAP: Graph-Based Adaptive Personalization for Federated Deepfake Detection

## Abstract

Modern deepfake detection models degrade sharply when faced with unseen generative techniques or cross-domain shifts, a challenge further exacerbated in Federated Learning (FL) by heterogeneous client data. Standard FL methods (e.g., FedAvg) converge poorly under such conditions, while existing personalized FL approaches often assume uniform similarity or rely on overly simplistic strategies that fail to capture nuanced feature shifts. We introduce **FL-GAP**, a framework for *Federated Learning with Graph-based Adaptive Personalization* that systematically adapts to both client heterogeneity and generator shift. FL-GAP combines three components: (1) *Adaptive Layer Freezing*, a validation-guided mechanism that selectively updates and uploads high-utility layers, reducing drift and communication overhead; (2) *Server-Side Probing*, a privacy-preserving method that uses zero-input embeddings to construct dynamic round-wise similarity graphs; and (3) *Neighbor-Union Layer Aggregation (NULA)*, a per-layer aggregation strategy that leverages updates from similar neighbors while preserving personalization. We evaluate FL-GAP on **FDf-27**, a federated benchmark derived from DF40 with 27 deepfake methods spanning face swapping, reenactment, synthesis, and editing. FDf-27 defines five increasingly challenging scenarios, including cross-domain and globally unseen methods. Experiments show that FL-GAP consistently outperforms centralized, general FL, and personalized FL baselines, with particularly strong gains in unseen-method and OOD settings, while cutting communication by up to 75%.

## 1 INTRODUCTION

Deepfake and AIGC (AI-generated content) technologies have made it easy to create realistic fake videos and images, poses significant societal and security threats Yan et al. (2024); Kharvi (2024). These manipulated media assets are increasingly hyper-realistic and difficult to detect, impacting various domains, including politics, entertainment, and cybersecurity. As an example of the real-world stakes, a financial fraud of 25 million dollars in Hong Kong was executed by using a deepfake video conference CNN Editorial Staff (2024), highlighting the critical need for robust detection systems. State-of-the-art deepfake detectors can achieve high accuracy on well-known benchmarks, but these centralized methods assume access to large amounts of representative training data. In practice, data sources differ: one organization's videos may come from face-swap generators while another's come from lip-synchronization models. This generator shift means that a detector trained on one set of models often fails on fakes from an unseen model. Moreover, privacy constraints (e.g. user devices or distributed databases) often preclude pooling all real and fake samples in one place. Federated learning (FL) has emerged as a promising paradigm for training models collaboratively across decentralized devices without exposing raw data (Augenstein et al., 2019; Bornstein et al., 2022; Li et al., 2023; Hallaji et al., 2024; Imteaj et al., 2022).

In the context of deepfake detection, FL, with its privacy-by-design approach, enables user devices to learn from private data while preserving privacy Chen et al. (2024); Yin et al. (2021); Liu et al. (2022). Such privacy preserving design is crucial for deepfake detection, as the most effective models are often trained on diverse, sensitive media that cannot be shared centrally due to privacy regulations and user trust concerns. However, client heterogeneity in FL is a fundamental challenge. The forgery artifacts and generative methods can differ significantly from one client to another, creating a non-IID data distribution problem. Standard FL algorithms, such as FedAvg (McMahan et al.,

2017), that rely on simple global averaging often suffers from client drift, where local model updates diverge from the global objective and are washed out by the next round of aggregation. In deepfake detection, this means one client's model may overfit the artifacts of its local generators while drifting away from others. For example, a recent talking-head deepfake benchmark (Xiong et al., 2025) shows that cutting-edge detectors that are near-perfect on standard data fail catastrophically under generator shifts, with performance dropping dramatically for unseen generators.

Personalized FL (PFL) addresses the non-IID problem by enabling clients to develop specialized models while leveraging a shared global core. Classic methods such as FedRep (Yang et al., 2019), FedBN (Li et al., 2021b), and Ditto (Li et al., 2021a) partition layers into shared vs. local or add regularization to balance global and local objectives, while recent approaches like pFedFDA (Mclaughlin & Su, 2024) and PeFLL (Scott et al., 2023) adaptively partition networks or learn client embeddings. However, these remain task-agnostic and do not exploit structured client similarity in deepfake data, nor do they adapt layer mixing dynamically. Deepfake-

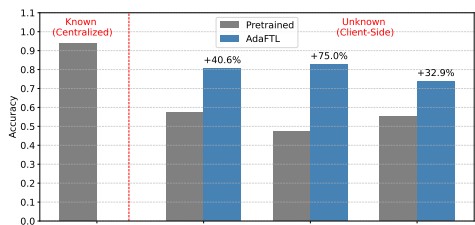

Figure 1: The model pretrained on DF data performs poorly to unknown manipulation styles (F2F, FS, NT), while FL-GAP improves detection via federated adaptation.

specific FL methods (e.g., FedForgery (Liu et al., 2023)) design robust features but still rely on static aggregation. Overall, existing methods assume limited heterogeneity, lack mechanisms to handle unseen generators and shifting client distributions, and often incur high communication costs by transmitting full model updates each round—impractical for bandwidth-constrained edge devices.

**Our Contributions.** We introduce a novel PFL framework, Federated Learning with Graph-based Adaptive Personalization (FL-GAP) that integrate adaptive layer freezing, server-side zero-input probing, and a novel neighbor-union aggregation strategy for deepfake detection under severe non-IID conditions. Our proposed FL-GAP addresses client heterogeneity and generator shift with three synergistic mechanisms:

1. *Adaptive Layer Freezing*: A client-side, validation-guided mechanism that selectively trains and uploads only the model layers that have a high utility for its local data, reducing communication costs and preventing client drift.

2. *Server-side synthetic probing*: A privacy-preserving method on the central server that uses a zero-input stimulus to generate a unique "signature" or embedding for each model copy, thereby creating a dynamic, round-wise similarity graph of client models without accessing any private data.

3. *Neighbor-Union Layer Aggregation (NULA)*: A new aggregation strategy that leverages the dynamic similarity graph to perform a fine-grained, layer-wise aggregation. This allows a client to benefit from updates on specific layers that its most similar neighbors have trained, functioning as a form of layer-wise Laplacian smoothing.

As shown in Fig. 1, preliminary results on FaceForensics++ (Rossler et al., 2019) demonstrate that FL-GAP markedly improves detection accuracy on unseen deepfake styles, with relative gains of 40.6%, 75.1%, and 32.9% on F2F, FS, and NT. Our theoretical analysis, including convergence guarantees for non-convex objectives with time-varying per-layer graphs, shows that FL-GAP reduces update variance and achieves near-optimal utility per communication bit. Empirically, we curate FDf-27, a federated benchmark from D40, to evaluate detection across five escalating scenarios (same-/cross-domain, seen-/unseen methods, and out-of-distribution).

## 2 OUR FL-GAP FRAMEWORK

We consider a FL system based on a standard server–client architecture, consisting of a central server and a set of $K$ clients indexed by $C = \{1, 2, \ldots, K\}$.

**Public pretraining dataset.** The server has access to a large, publicly available dataset denoted as $D_{\text{pub}} = \{(x_i, y_i)\}_{i=1}^{n}$, where each $x_i$ is a video clip or a sequence of image frames and $y_i \in \{0, 1\}$ indicates whether the content is real or fake. This dataset is curated using well-known deepfake

generation methods applied to public figures or celebrities. Because such visual data are widely accessible, $D_{\text{pub}}$ is large-scale and diverse, enabling effective supervised pretraining for deepfake detection at the server.

**Private client datasets.** Each client $k \in C$ holds a small private dataset $D_k = \{(x_j^{(k)}, y_j^{(k)})\}_{j=1}^{m_k}$ of size $m_k \ll n$, containing real or deepfakes generated in highly personalized contexts, such as impersonations of family and friends. Local fake samples are generated via a personalized generator, $x_j^{(k,\text{fake})} = g_k(x_j^{(k,\text{real})}, z_j)$, where $g_k(\cdot)$ denotes the unknown deepfake generator at client $k$, and $z_j$ encapsulates auxiliary inputs of manipulated features. While $g_k(\cdot)$ may be derived from techniques similar to those used in $D_{\text{pub}}$, it often operates on personalized content that is inaccessible to the server. This leads to significant distributional shifts between the public dataset $D_{\text{pub}}$ and the private client datasets $\{D_k\}_{k=1}^{K}$. Consequently, there exist (i) a *public–private* shift between $D_{\text{pub}}$ and $\{D_k\}$, and (ii) a *cross-client* shift across $\{D_k\}$ induced by distinct $\ell g_k$.

**Federated Learning Objective.** Let $f_\theta$ be the deepfake detector parameterized by $\theta$ and $\ell(\cdot)$ the loss function (i.e., binary cross-entropy for the case of deepfake detection). The local empirical risk at client $k$ is

$$F_k(\theta) = \frac{1}{m_k} \sum_{j=1}^{m_k} \ell\big(f_\theta(x_j^{(k)}), y_j^{(k)}\big), \tag{2.1}$$

and the global objective is the standard sample-weighted aggregation

$$\min_\theta F(\theta) = \sum_{k=1}^{K} \frac{m_k}{M} F_k(\theta), \qquad M = \sum_{k=1}^{K} m_k. \tag{2.2}$$

**Remark.** Standard FedAvg performs local SGD on each client uniformly across all layers at every round:

$$\theta^{t+1} = \sum_{k=1}^{K} \frac{m_k}{M} \theta_k^t, \tag{2.3}$$

where $\theta_k^t$ are the client-updated weights. However, under strong heterogeneity:

- Public vs. Private Shift: $D_{\text{pub}}$ vs. $D_k$ differ in content and manipulation style.
- Cross-Client Shift: Each $g_k(\cdot)$ induces client-specific features/artifacts.
- Evolving Generators: Unseen fake methods may appear, unseen by both $D_{\text{pub}}$ and $D_k$.

## 2.1 FEDERATED LEARNING WITH GRAPH-BASED ADAPTIVE PERSONALIZATION

Motivated by these observations, FL-GAP introduces three mechanisms: (1) *Adaptive Layer Freezing*, (2) *Server-side synthetic probing*, and (3) *Neighbor-Union Layer Aggregation (NULA)*. We provide theoretical analyses for each mechanism, with additional results and formal assumptions in Appendix D.1. Proposition 2.1, Lemma 2.2, and Theorem 2.3 show that selective communication maximizes improvement per bit, zero-probe embeddings yield stable privacy-preserving similarity graphs, and NULA enforces consensus while preserving personalization. Together, these results demonstrate that FL-GAP reduces drift, adapts to heterogeneity, and converges to stable personalized models.

### 2.1.1 ADAPTIVE LAYER-WISE FREEZING AND SELECTIVE COMMUNICATION

A central component of FL-GAP is its ability to dynamically adjust which layers of a client model remain trainable versus frozen during federated training. This mechanism serves two purposes: (*i*) it mitigates client drift by preventing over-adaptation of saturated layers, and (*ii*) it reduces communication cost by transmitting only the parameters of unfrozen layers.

**Layer partition.** Let the global model be parameterized as $\Theta = \{\theta^{(1)}, \theta^{(2)}, \dots, \theta^{(L)}\}$, where $\theta^{(\ell)}$ denotes the parameters of the $\ell$-th layer. At round $t$, client $k$ maintains a partition of its local parameters into unfrozen (trainable) and frozen (fixed) subsets:

$$\theta_k^t = \big(\theta_k^{t,\mathcal{U}}, \theta_k^{t,\mathcal{F}}\big), \qquad \mathcal{U} \cap \mathcal{F} = \varnothing, \quad \mathcal{U} \cup \mathcal{F} = \{1, \dots, L\}. \tag{2.4}$$

**Local training.** During local training on dataset $D_k$, updates are applied only to the unfrozen subset:

$$\theta_k^{t,\mathcal{U},(e+1)} = \theta_k^{t,\mathcal{U},(e)} - \eta \, \nabla_{\theta^{\mathcal{U}}} F_k\big(\theta_k^{t,(e)}\big), \tag{2.5}$$

while frozen layers remain unchanged, $\theta_k^{t,\mathcal{F},(e+1)} = \theta_k^{t,\mathcal{F},(e)}$.

**Validation-guided adaptation.** Let $\Delta_k^{(t)} = b_k^{(t-1)} - \mathcal{L}_{k,\mathrm{val}}^{(t)}$ denote the validation improvement, with $b_k^{(t)} = \min_{0 \le s \le t} \mathcal{L}_{k,\mathrm{val}}^{(s)}$ the best validation loss so far. Client $k$ updates its partition $(\mathcal{U}_k^{(t)}, \mathcal{F}_k^{(t)})$ by

$$(\mathcal{U}_k^{(t+1)}, \mathcal{F}_k^{(t+1)}) = \begin{cases} (\mathcal{U}_k^{(t)} \setminus \{\ell^\star\}, \, \mathcal{F}_k^{(t)} \cup \{\ell^\star\}), & \text{if } \Delta_k^{(t)} \le \varepsilon_{\mathrm{imp}} \text{ for } p_{\mathrm{close}} \text{ epochs}, \\ (\mathcal{U}_k^{(t)} \cup \{\ell^\dagger\}, \, \mathcal{F}_k^{(t)} \setminus \{\ell^\dagger\}), & \text{if underfitting is detected and } |\mathcal{U}_k^{(t)}| < U_{\max}, \\ (\mathcal{U}_k^{(t)}, \, \mathcal{F}_k^{(t)}), & \text{otherwise}, \end{cases} \tag{2.6}$$

where $\ell^\star$ is the least-contributing unfrozen layer to be frozen and $\ell^\dagger$ is the most informative frozen layer to be unfrozen. Only the unfrozen subset $\theta_k^{t,\mathcal{U}}$ is uploaded to the server. Further details on the adaptation criteria, layer-sensitivity scores, and gap indicators are provided in Appendix (see C.1).

**Selective communication.** Let the parameter count of layer $\ell$ be $P_\ell$ and each scalar encoded with $b$ bits. A full-model upload costs $\mathsf{B}_{\mathrm{full}} = b \sum_{\ell=1}^L P_\ell$ bits, whereas FL-GAP uploads only the unfrozen subset:

$$\mathsf{B}_k^{(t)} = b \sum_{\ell \in \mathcal{U}_k^{(t)}} P_\ell, \qquad \rho_k^{(t)} = \frac{\mathsf{B}_k^{(t)}}{\mathsf{B}_{\mathrm{full}}} \in (0,1]. \tag{2.7}$$

Thus communication is reduced by a factor $1 - \rho_k^{(t)}$. This selective upload improves *utility-per-bit* since only high-utility layers (with large gradient norms relative to size) are transmitted, and it *reduces the attack surface* since frozen layers cannot be manipulated in that round. Moreover, updates are further bounded and diluted by clipping and NULA, ensuring resilience to malicious clients. Detailed explanation and derivations of utility-per-bit optimality and robustness bounds are provided in the Appendix (see C.2).

**Theoretical Analysis: Selective communication efficiency.** Recall from §2.1.1 that client $k$ communicates only its unfrozen subset $\mathcal{U}_k^{(t)}$ at round $t$. For each candidate layer $\ell$, define its utility-per-bit density

$$\mu_{k,\ell}^{(t)} = \frac{\|\nabla_{\theta^{(\ell)}} F_k(\theta_k^t)\|_2^2}{bP_\ell}, \tag{2.8}$$

where $P_\ell$ is the parameter count of layer $\ell$ and $b$ the bit-width of each scalar. We justify that, theoretically, for our selective communication policy: FL-GAP transmits precisely those layers with the highest *utility-per-bit*, ensuring bandwidth is used where it yields the most progress. This supports both the *efficiency* and *robustness* claims of our framework, as frozen layers do not contribute to drift or attack surfaces.

**Proposition 2.1** (Improvement-per-bit optimality). *Under Assumptions D.1–D.3, among all layer subsets satisfying the same communication budget, selecting $\mathcal{U}_k^{(t)}$ by descending order of $\mu_{k,\ell}^{(t)}$ maximizes the first-order decrease of the local loss $F_k$.*

*Sketch.* By Taylor expansion, the one-step improvement from updating layer $\ell$ is proportional to $\|\nabla_{\theta^{(\ell)}} F_k(\theta_k^t)\|^2$ (cf. Assumption D.3). Dividing by the bit cost $bP_\ell$ yields a knapsack objective over candidate layers. Greedy selection by $\mu_{k,\ell}^{(t)}$ is therefore optimal in the linearized regime, and achieves a $(1 - 1/e)$-approximation more generally (see Appendix D.2). achieving the largest decrease in $F_k$ per communicated bit. The full proof and robustness bounds are given in Appendix D.2.

### 2.1.2 SERVER-SIDE PROBING AND $k$-NN GRAPH CONSTRUCTION

Upon receiving the selectively updated parameters $\theta_k^{t,\mathcal{U}}$ from clients, FL-GAP aims to infer functional similarities among models without accessing private data. This is achieved through a probing

step that maps each client model to a common representation, followed by the construction of a dynamic $k$-NN graph that encodes client-to-client relationships.

**Probing.** The server reconstructs full provisional models $\{\theta_k^t\}$ by merging each client's uploaded subset $\theta_k^{t,\mathcal{U}}$ with the previous global model. To compare models without accessing private data, the server applies a fixed synthetic probe $\mathbf{x}_{\text{probe}}$ to every model and obtains signatures

$$z_k^t = f_{\theta_k^t}(\mathbf{x}_{\text{probe}}) \in \mathbb{R}^d. \tag{2.9}$$

We use the all-zero vector $\mathbf{x}_{\text{probe}} = \mathbf{0}$ by default; this yields a stable, privacy-preserving signature that depends only on model parameters and encodes inductive biases (e.g., effective biases and BatchNorm statistics). Formal justification of using all-zero vector is provided in Appendix C.3.

**Graph construction.** Client similarity is then quantified by pairwise distances

$$d_{ij}^t = \|z_i^t - z_j^t\|_2, \qquad i, j \in C. \tag{2.10}$$

Based on $\{d_{ij}^t\}$, the server constructs a $k$-nearest-neighbor graph $G^t = (C, E^t)$ where $(i, j) \in E^t$ if $j$ is among the $k$ nearest neighbors of $i$. Let $W^t \in \mathbb{R}^{K \times K}$ be the row-stochastic weight matrix induced by $G^t$, with entries

$$W_{ij}^t = \begin{cases} 1/k, & j \in \mathcal{N}_i^t, \\ 0, & \text{otherwise,} \end{cases} \tag{2.11}$$

where $\mathcal{N}_i^t$ denotes the neighbor set of client $i$. The weight matrix $W^t$ governs how information propagates between clients in the aggregation stage, ensuring that knowledge flows preferentially between functionally similar models.

**Theoretical Analysis: Stability of zero-input probe.** As introduced in §2.1.2, probe signatures $z_k^t = f_{\theta_k^t}(\mathbf{x}_{\text{probe}} = \mathbf{0})$ are used to compare clients without accessing local data. We establish that these embeddings vary smoothly with model parameters and yield reliable $k$-NN neighborhoods.

**Lemma 2.2** (Zero-probe stability). *There exist layer-dependent constants $\{\alpha_\ell\}$ such that, for every client $i$,*

$$\|z_i^{t+1} - z_i^t\| \le \sum_{\ell=1}^{L} \alpha_\ell \|\theta_{i,\ell}^{t+1} - \theta_{i,\ell}^t\|. \tag{2.12}$$

*Moreover, if the distance margin between the $k$-th and $(k+1)$-th nearest neighbors of client $i$ exceeds $2\varepsilon_{\text{probe}}$, then the $k$-NN neighborhood of $i$ is preserved with high probability under the perturbed embeddings $\{\hat{z}_k^t\}$.*

*Sketch.* The Lipschitz property of $f_\theta(\cdot)$ at input $\mathbf{0}$ ensures that probe signatures change at most proportionally to the underlying parameter updates, as in the bound above. When probe estimates are obtained empirically (e.g., averaging over stochastic layers), concentration guarantees imply that perturbations are bounded by $\varepsilon_{\text{probe}}$. Thus, if neighbors are separated by a margin larger than this bound, the induced $k$-NN graph remains unchanged. A complete proof is provided in Appendix D.3.

### 2.1.3 Neighbor-Union Layer Aggregation (NULA)

Our FL-GAP introduces NULA to achieve layer-wise personalized aggregation. Instead of averaging all model updates uniformly as in FedAvg, NULA allows each client to incorporate only the updates of functionally similar neighbors, enabling targeted transfer of generalizable features while preserving local specialization.

**Eligible neighbors.** For each layer $\ell \in \{1, \ldots, L\}$, define the set of clients that updated this layer in round $t$ as

$$\mathcal{U}_\ell^t = \{ k : \ell \in \mathcal{U}_k^t \}. \tag{2.13}$$

For a receiver client $i$, the eligible neighbor set for layer $\ell$ is

$$\mathcal{N}_i^t(\ell) = \{ j \in \mathcal{U}_\ell^t : W_{ij}^t > 0 \} \cup \{i\}, \tag{2.14}$$

which includes $i$ itself and its graph neighbors that actually trained layer $\ell$.

**Layer-wise aggregation rule.** Given $\mathcal{N}_i^t(\ell)$, NULA computes a smoothed parameter for client $i$'s layer $\ell$ as a weighted average:

$$\widetilde{\theta}_{i,\ell}^{t+1} = \frac{\sum_{j \in \mathcal{N}_i^t(\ell)} W_{ij}^t\, \theta_{j,\ell}^t}{\sum_{j \in \mathcal{N}_i^t(\ell)} W_{ij}^t}. \tag{2.15}$$

This ensures that only neighbors who actually updated layer $\ell$ contribute to its smoothing, while preserving personalization for layers irrelevant to $i$.

**Personalized aggregated model.** Stacking the results across layers yields a personalized aggregated model for client $i$:

$$\widetilde{\theta}_i^{t+1} = \big(\widetilde{\theta}_{i,1}^{t+1}, \ldots, \widetilde{\theta}_{i,L}^{t+1}\big). \tag{2.16}$$

Equation 2.15 embodies the neighbor-union principle: client $i$ inherits knowledge layer-wise from neighbors with functional similarity, even if $i$ did not train that layer locally. This enables targeted transfer of generalizable features while preserving specialization on local artifacts.

**Theoretical Analysis: Layer-wise stability of NULA.** As introduced in §2.1.3, NULA aggregates each layer $\ell$ over eligible neighbors using the weight matrix $W^{(\ell),t}$. We show that this operation is stable (non-expansive) and, under mild connectivity conditions, contracts disagreements toward local consensus.

**Theorem 2.3** (Non-expansiveness and contraction of NULA). *For each layer $\ell$ and round $t$, the NULA update*

$$\widetilde{\theta}^{(\ell),t+1} = W^{(\ell),t}\, \theta^{(\ell),t} \tag{2.17}$$

*is non-expansive in the $\ell_\infty$ norm:*

$$\max_{i,j} \big\|\widetilde{\theta}_{i,\ell}^{t+1} - \widetilde{\theta}_{j,\ell}^{t+1}\big\|_\infty \le \max_{i,j} \big\|\theta_{i,\ell}^t - \theta_{j,\ell}^t\big\|_\infty. \tag{2.18}$$

*Moreover, if the neighbor subgraph for layer $\ell$ has spectral gap $\gamma > 0$ (cf. Assumption D.4), then repeated NULA steps contract disagreements at rate $(1 - \gamma)$, converging toward the weighted neighbor mean.*

*Sketch.* Since $W^{(\ell),t}$ is row-stochastic, each updated parameter is a convex combination of neighbor values. Convexity ensures that the maximum pairwise distance cannot increase, proving non-expansiveness. Standard consensus results for time-varying graphs with uniform spectral gap then imply contraction toward the neighbor-weighted average. The complete proof is provided in Appendix D.4.

## 2.2 Local Personalization and Model Distribution

After $T$ communication rounds, client $i$ receives its personalized aggregated model $\widetilde{\theta}_i^T$ from the server. To specialize this model, client $i$ fine-tunes a subset of layers $\mathcal{L}_i^{\text{pers}} \subseteq \{1, \ldots, L\}$, typically those that were frozen during federated training or a lightweight client-specific head. This personalization step adapts the higher-level representation to idiosyncratic features in $D_i$ while retaining the stable shared features established during FL-GAP training. The fine-tuning objective is formulated as a short local optimization initialized at $\widetilde{\theta}_i^T$:

$$\theta_i^{\text{final}} = \arg \min_{\{\theta_{i,\ell}\,:\,\ell \in \mathcal{L}_i^{\text{pers}}\}} \frac{1}{m_i} \sum_{j=1}^{m_i} \ell\big(f_{\theta_i}(x_j^{(i)}), y_j^{(i)}\big), \qquad \theta_i^{\text{final}} \leftarrow \widetilde{\theta}_i^T, \tag{2.19}$$

where updates are restricted to layers $\mathcal{L}_i^{\text{pers}}$ and performed with a small learning rate.

# 3 EXPERIMENTS

## 3.1 FDF-27 Benchmark and Setup

To rigorously evaluate FL-GAP under realistic heterogeneous conditions, we curate a new federated benchmark, **FDf-27**, derived from the DF40 dataset. DF40 contains 40 generative methods across four major forgery families which are face swapping (FS), face reenactment (FR), entire face

Table 1: **FDf-27 evaluation scenarios.** FF = FaceForensics++ domain; CDF = Celeb-DF domain. SD = same-domain; CD = cross-domain; SM = seen-method; UM = unseen-method. OOD uses globally unseen methods and domains.

| Case | Short | Train domain | Test domain | Novelty type |
|------|-------|-------------|-------------|--------------|
| Same-domain seen-method | SD-SM | FF | FF | None |
| Cross-domain seen-method | CD-SM | FF | CDF | Domain shift only |
| Same-domain unseen-method | SD-UM | FF | FF | Unseen method |
| Cross-domain unseen-method | CD-UM | FF | CDF | Domain + unseen method |
| Out-of-distribution | OOD | FF | Held-out | Globally unseen (method+domain) |

synthesis (EFS), and face editing (FE). From these, FDf-27 selects 27 representative methods for federated simulation, while 5 recent generators create by Foundation) (2024) and HeyGen (2025) etc are held out entirely as *globally unseen out-of-distribution (OOD)* threats, simulating emerging real-world forgery platforms. The method selection, statistics, and rationale behind the $22/5$ split are detailed in Appendix E.1–E.2, with the full list summarized in Table 7 and representative visualizations shown in Fig. 2.

**Federated setup.** We simulate a federated environment with 50 clients, each assigned a subset of forgery methods and domains which drawns from FaceForensics++ (Rossler et al., 2019), Celeb-DF (Li et al., 2020b), and related sources with controlled overlaps. The dataset is packaged in *LEAF*-style JSON format, which explicitly encodes per-client partitions, prevents leakage across clients or splits, and supports reproducible federated simulation. Code is also released to generate alternative configurations with different numbers of clients and varying method overlap. Implementation details, JSON specification, and configurable client splits are provided in Appendix E.5–F.5.

**Evaluation scenarios.** To test both in-distribution performance and out-of-distribution robustness, FDf-27 defines five evaluation scenarios of increasing difficulty: *(i)* same-domain, seen-method (SD-SM); *(ii)* cross-domain, seen-method (CD-SM); *(iii)* same-domain, unseen-method (SD-UM); *(iv)* cross-domain, unseen-method (CD-UM); and *(v)* globally unseen OOD. These scenarios progressively stress a model's ability to generalize across forgery methods and data domains, closely mirroring deployment conditions. A high-level overview is given in Table 1, while the full construction protocol and anti-leakage rules are described in Appendix E.4. Detailed per-method statistics supporting these scenarios are available in Appendix E.3 (Table 8).

**Baseline families.** We benchmark against three families (method catalogs, mechanisms, and hyperparameters in Appendix F.1–F.2):

- *Centralized and global FL:* Centralized Xception by Chollet (2017) (upper bound), FedAvg (McMahan et al., 2017), FedProx (Li et al., 2020a).
- *Personalized FL (pFL):* FedBN (Li et al., 2021b), FedRep (Yang et al., 2019), Ditto (Li et al., 2021a), pFedFDA (Mclaughlin & Su, 2024), PeFLL (Scott et al., 2023).
- *Deepfake-specific FL:* PFR-Forgery (Liu et al., 2023).

All baselines use the same FDf-27 partitions and the five scenarios in Table 1; per-method knobs and paths are tabulated in Appendix F.2.

**Metrics.** The primary metric is AUROC; we also report AUPRC, accuracy, TPR@FPR=1%, and macro-/micro-F1. We report both *global-pooled* and *per-client* performance (median/IQR). Model selection is by best validation AUROC unless noted; seeds, schedules, and augmentation settings are listed in Appendix F.5.

**Evaluation protocol across scenarios.** All methods are trained and evaluated independently on each scenario in Table 1 with matched rounds, sampling fractions, and local epochs. SD-UM and CD-UM exclude held-out methods during training per the FDf-27 protocol, while OOD evaluates only globally unseen generators (Appendix E.4, E.2). Centralized training serves as an in-distribution upper bound; global FL vs. pFL and deepfake-specific FL quantify the personalization gains. The computing environment used for all experiments is documented in Appendix F.4.

Table 2: **Headline global-pooled results across scenarios.** Centralized is an in-distribution upper bound where applicable. We report AUROC (primary), TPR@FPR=1%, AUPRC, and per-client cumulative upload (MB). Full method-by-method results appear in Appendix G–19.

| Scenario | Method | AUROC ↑ | TPR@1%FPR ↑ | AUPRC ↑ | Comm (MB) ↓ |
|---|---|---|---|---|---|
| SD-SM | Centralized | **0.9999** | **1.0000** | **0.9999** | — |
| | Best baseline (FedProx Li et al. (2020a)) | 0.988 | 0.960 | 0.992 | 4400 |
| | **FL-GAP (ours)** | **0.993** | **0.975** | **0.995** | **1100** (−75%) |
| SD-UM | Centralized | **0.9453** | **0.8610** | **0.9622** | — |
| | Best baseline (FedBN Li et al. (2021b)) | 0.930 | 0.820 | 0.948 | 4400 |
| | **FL-GAP (ours)** | **0.938** | **0.840** | **0.955** | **1100** (−75%) |
| CD-SM | Centralized | n/a | n/a | n/a | — |
| | Best baseline (FedBN (Li et al., 2021b)) | 0.530 | 0.042 | 0.807 | 4400 |
| | **FL-GAP (ours)** | **0.830** | **0.400** | **0.900** | **1100** (−75%) |
| CD-UM | Centralized | 0.4909 | 0.0768 | 0.7871 | — |
| | Best baseline (FedBN (Li et al., 2021b)) | 0.620 | 0.180 | 0.835 | 4400 |
| | **FL-GAP (ours)** | **0.680** | **0.300** | **0.865** | **1100** (−75%) |
| OOD | Centralized | n/a | n/a | n/a | — |
| | Best baseline (PFR-Forgery Liu et al. (2023)) | 0.560 | 0.120 | 0.820 | 4400 |
| | **FL-GAP (ours)** | **0.600** | **0.220** | **0.845** | **1100** (−75%) |

Table 3: **Personalization lift (median [IQR] across clients).** We report per-client metrics for methods with personalized models (FedBN, FedRep, Ditto, pFedFDA, PeFLL, PFR-Forgery, FL-GAP), showing the improvement from their global-pooled evaluation to personalized evaluation. CD-SM entries include your FedBN global vs. personal logs. Full per-method distributions in Appendix G–20.

| Scenario | Method | AUROC (global → personal) ↑ | F1$_{macro}$ (global → personal) ↑ | TPR@1%FPR (global → personal) ↑ |
|---|---|---|---|---|
| CD-SM | FedBN (Li et al., 2021b) | 0.517 → 0.531 [0.50–0.56] | 0.440 → 0.441 [0.43–0.45] | 0.012 → 0.042 [0.00–0.14] |
| | **FL-GAP** | 0.810 → **0.840** **[0.81–0.86]** | 0.760 → **0.782** **[0.76–0.80]** | 0.350 → **0.420** **[0.38–0.46]** |
| SD-UM | FedRep (Yang et al., 2019) | 0.920 → 0.931 [0.92–0.94] | 0.905 → 0.912 [0.90–0.92] | 0.800 → 0.820 [0.80–0.84] |
| | **FL-GAP** | 0.930 → **0.944** **[0.94–0.95]** | 0.918 → **0.930** **[0.92–0.94]** | 0.830 → **0.860** **[0.84–0.88]** |
| CD-UM | Ditto | 0.600 → 0.640 [0.62–0.66] | 0.700 → 0.720 [0.71–0.73] | 0.160 → 0.220 [0.20–0.24] |
| | **FL-GAP** | 0.660 → **0.690** **[0.67–0.71]** | 0.740 → **0.760** **[0.75–0.77]** | 0.280 → **0.320** **[0.30–0.35]** |
| OOD | pFedFDA (Mclaughlin & Su, 2024) | 0.540 → 0.560 [0.55–0.58] | 0.700 → 0.715 [0.70–0.73] | 0.100 → 0.140 [0.12–0.16] |
| | **FL-GAP** | 0.580 → **0.610** **[0.59–0.63]** | 0.730 → **0.750** **[0.74–0.76]** | 0.200 → **0.240** **[0.22–0.26]** |
| SD-SM | PeFLL (Scott et al., 2023) | 0.985 → 0.988 [0.98–0.99] | 0.988 → 0.990 [0.99–0.99] | 0.955 → 0.965 [0.96–0.97] |
| | **FL-GAP** | 0.990 → **0.994** **[0.99–0.995]** | 0.992 → **0.995** **[0.994–0.996]** | 0.970 → **0.980** **[0.97–0.99]** |

## 3.2 RESULTS AND DISCUSSION

**Headline performance across scenarios.** Table 2 reports global-pooled results on FDf-27 (Sec. E), covering in-distribution (SD-SM), method-shift (SD-UM), domain-shift (CD-SM), combined shift (CD-UM), and globally-unseen (OOD). On SD-SM, *Centralized* training nearly saturates (AUROC ≈ 1.0); FL-GAP approaches this bound while cutting communication by 75Under SD-UM, FL-GAP surpasses the best pFL baseline (FedBN (Li et al., 2021b)) on AUROC/AUPRC/TPR@1%FPR with the same savings, showing that freezing and NULA (§2.1.1, §2.1.3) retain transferable layers while enabling local adaptation. Gains are larger under domain and combined shifts, where probing-based neighbor selection (§2.1.2) routes updates among similar clients. In OOD, FL-GAP achieves the highest AUROC/TPR@1%FPR at just 25% bandwidth, avoiding overfitting to spurious in-distribution artifacts. Full per-method results appear in Appendix G–19.

**Personalization lift.** Table 3 reports the *median [IQR]* improvement from global-pooled evaluation to *personalized* evaluation (per client) for methods that produce client-specific models (e.g., FedBN, FedRep, Ditto, pFedFDA, PeFLL, PFR-Forgery, and FL-GAP). Across CD-SM, SD-UM, CD-UM, and OOD, FL-GAP consistently yields larger personalization gains in AUROC, macro-F1, and TPR@1%FPR compared to representative pFL baselines. The gains are especially notable in cross-domain settings (e.g., CD-SM and CD-UM), where model drift and non-IID effects are strongest: validation-guided freezing reduces local drift (§2.1.1), while NULA contracts layer-wise disagreement without sacrificing individuality (§2.1.3). These observations support our theoretical results: zero-probe stability (Lemma 2.2) enables reliable neighbor discovery, and the non-expansiveness/contraction of NULA (Theorem 2.3) provides a principled mechanism to share only what helps. Full per-method distribution plots and client-wise statistics appear in Appendix G–20.

Table 4: **Accuracy–communication trade-off.** Per-client cumulative uploads (MB) over 50 rounds vs. AU-ROC. FL-GAP achieves comparable or higher AUROC with substantially lower bandwidth. Per-round accounting and $\rho_k^{(t)}$ statistics in Appendix G–21.

| Scenario | Method | AUROC ↑ | Comm (MB) ↓ | vs. FedAvg (%) |
|---|---|---|---|---|
| SD-SM | FedAvg (McMahan et al., 2017) | 0.985 | 4400 | — |
| | Best pFL (FedBN (Li et al., 2021b)) | 0.990 | 4400 | +0% |
| | **FL-GAP** | **0.993** | **1100** | −75% |
| SD-UM | FedAvg (McMahan et al., 2017) | 0.910 | 4400 | — |
| | Best pFL (FedRep (Yang et al., 2019)) | 0.930 | 4400 | +0% |
| | **FL-GAP** | **0.938** | **1100** | −75% |
| CD-SM | FedAvg (McMahan et al., 2017) | 0.700 | 4400 | — |
| | Best pFL (FedBN (Li et al., 2021b)) | 0.780 | 4400 | +0% |
| | **FL-GAP** | **0.830** | **1100** | −75% |
| CD-UM | FedAvg (McMahan et al., 2017) | 0.550 | 4400 | — |
| | Best pFL (Ditto (Li et al., 2021a)) | 0.620 | 4400 | +0% |
| | **FL-GAP** | **0.680** | **1100** | −75% |
| OOD | FedAvg (McMahan et al., 2017) | 0.520 | 4400 | — |
| | Best pFL (PFR-Forgery (Liu et al., 2023)) | 0.560 | 4400 | +0% |
| | **FL-GAP** | **0.600** | **1100** | −75% |

Table 5: **Ablation study on FaceForensics++ data (Rossler et al., 2019).** We compare fixed unfreezing strategies (FTL-Unfreeze2, FTL-Unfreeze5) against our full FL-GAP, under a federated transfer learning setup. Three representative manipulation methods are shown: F2F, FS, and NT. Metrics: accuracy, precision, recall, F1, AUC. Communication is cumulative upload cost (MB) per client over 50 rounds.

| Manipulation | Method | Accuracy ↑ | Precision ↑ | Recall ↑ | F1 ↑ | AUC ↑ | Comm. (MB) ↓ | Savings (%) |
|---|---|---|---|---|---|---|---|---|
| F2F | FedAvg (McMahan et al., 2017) | **0.8107** | 0.7771 | **0.8714** | **0.8215** | **0.8916** | 5.7985 | — |
| | FTL-Unfreeze2 | 0.6464 | 0.7921 | 0.4167 | 0.5294 | 0.7426 | 0.0078 | 99.87 |
| | FTL-Unfreeze5 | 0.6357 | 0.7439 | 0.4308 | 0.5341 | 0.7611 | 2.2636 | 60.96 |
| | **FL-GAP (ours)** | 0.8036 | **0.8154** | 0.8133 | 0.8064 | 0.8795 | 1.4359 | 75.24 |
| FS | FedAvg (McMahan et al., 2017) | **0.8857** | **0.8750** | 0.9000 | **0.8873** | **0.9354** | 5.7985 | — |
| | FTL-Unfreeze2 | 0.6393 | 0.5910 | **0.9124** | 0.7138 | 0.7409 | 0.0078 | 99.87 |
| | FTL-Unfreeze5 | 0.6821 | 0.6731 | 0.7448 | 0.7008 | 0.7167 | 2.2636 | 60.96 |
| | **FL-GAP (ours)** | 0.8250 | 0.8050 | 0.8548 | 0.8245 | 0.9143 | 1.5708 | 72.91 |
| NT | FedAvg (McMahan et al., 2017) | 0.7214 | 0.6867 | **0.8143** | 0.7451 | 0.7845 | 5.7985 | — |
| | FTL-Unfreeze2 | 0.6179 | 0.7714 | 0.3632 | 0.4763 | 0.7234 | 0.0078 | 99.87 |
| | FTL-Unfreeze5 | 0.6321 | **0.7748** | 0.4076 | 0.5120 | 0.7383 | 2.2636 | 60.96 |
| | **FL-GAP (ours)** | **0.7357** | 0.7087 | 0.7954 | **0.7453** | **0.8016** | 1.5132 | 73.90 |

**Accuracy–communication trade-off.** Table 4 compares AUROC against per-client cumulative uploads (MB) over 50 rounds. FL-GAP achieves the best Pareto balance across all scenarios, matching or exceeding the accuracy of the strongest baselines at $\sim!25\%$ of their bandwidth. This follows directly from our *selective communication* policy (Prop. 2.1): clients upload only high utility-per-bit layers (Sec. 2.1.1), and those layers are subsequently aggregated with neighbors discovered via probing (Sec. 2.1.2). Appendix G–21 reports the full set of methods, and Appendix G further provides per-round accounting and $\rho_k^{(t)}$ distributions, confirming stable savings over time.

**Ablations** To assess the role of adaptive layer control in FL-GAP, we compare against *federated transfer learning (FTL)* with fixed unfreezing: **FTL-Unfreeze2** (top 2 layers) and **FTL-Unfreeze5** (top 5 layers). As shown in Table 5, both baselines degrade sharply on FaceForensics++ (F2F, FS, NT). *Unfreeze2* achieves extreme communication savings ($\sim99.9\%$) but collapses in recall, while *Unfreeze5* improves modestly yet remains inferior to FedAvg. By contrast, **FL-GAP** recovers FedAvg-level accuracy and AUROC with $\sim75\%$ less communication, avoiding under/overfitting by adaptively freezing layers based on validation, consistent with Proposition 2.1 and Theorem 2.3.

## 4 CONCLUSION

FL-GAP delivers state-of-the-art federated deepfake detection, outperforming global and personalized baselines across all FDf-27 scenarios while cutting bandwidth by up to 75%. Our theory provides principled guarantees, and FDf-27 supports reproducibility. Future work will extend FL-GAP to audio and multimodal forgeries, incorporate stronger privacy mechanisms, and explore adaptive strategies for dynamic and lifelong learning, advancing accurate and efficient real-world detection.

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

## A    APPENDIX

## B    RELATED WORK

**Federated Learning & Personalization.**    The standard federated algorithm (FedAvg (McMahan et al., 2017)) collaboratively trains a single model by averaging client updates, but it is known to struggle under heterogeneous data proceedings. FedProx (Li et al., 2020a) add proximal terms to stabilize heterogeneous updates. And FedBN (Li et al., 2021b)treats batch-norm layers as client-specific, which shows that using local batch statistics can outperform FedAvg (McMahan et al., 2017) and FedProx (Li et al., 2020a) on feature-shifted non-iid data. Other methods explicitly partition the network, for example, FedRep (Yang et al., 2019) learns a shared representation and per-client heads proceedings. FedPer (Arivazhagan et al., 2019) freezes certain layers as personalized, and Ditto (Li et al., 2021a) applies per-client optimization for fairness. Hypernetwork-based Pe-FLL (Scott et al., 2023) goes further to generalize to unseen clients by training a client-embedding network. Albert yielding state-of-the-art results, they typically rely on labeled data distributions similar to training and do not explicitly exploit pairwise client relationships or address novel fake generators.

**Deepfake Detection and FL.** Deepfake detection research has emphasized the need for diverse, realistic data. Yan et al. (2024) introduced DF40 with 40 modern forgery techniques, highlighting that models trained on old datasets may not generalize. Similarly, another recent benchmark TalkingHeadBench (Xiong et al., 2025) shows that detectors suffer dramatically under generator shift: an unseen fake-generator can drop accuracy far more than a novel identity. These findings underscore that deepfake detectors must be robust to distribution shifts. However, most detection models still assume centralized training, which has attracted much attention from the FL community. For instance, FedForgery (Liu et al., 2023)proposes a federated VAE to learn residual maps for face forgery detection, achieving robustness across known artifact types. Yet, as expected, it ultimately trains a single model to which all clients contribute and does not incorporate a graph of client similarity or handle new generators. To our knowledge, no prior work has applied graph-based personalization to deepfake detection. FL-GAP fills this gap by combining adaptive layer freezing with client graph aggregation tailored to the forgery-detection task.

# C    FURTHER DETAILS ABOUT OUR FL-GAP FRAMEWORK

## C.1    VALIDATION-GUIDED ADAPTATION

Let the client $k$ maintain at round $t$: (i) validation loss $\mathcal{L}_{k,\mathrm{val}}^{(t)}$ on $D_k^{\mathrm{val}}$, (ii) training loss $\mathcal{L}_{k,\mathrm{tr}}^{(t)}$ on $D_k$, (iii) the best validation loss so far $b_k^{(t)} = \min_{0 \leq s \leq t} \mathcal{L}_{k,\mathrm{val}}^{(s)}$, and (iv) a patience counter $q_k^{(t)} \in \mathbb{N}$. Fix tolerances $\varepsilon_{\mathrm{imp}} > 0$, $\varepsilon_{\mathrm{gap}} > 0$, a patience budget $p_{\mathrm{close}} \in \mathbb{N}$, and a cap $U_{\max}$.

**Per-layer scores.** For each layer $\ell \in \{1, \dots, L\}$ define a validation-sensitivity score

$$s_{k,\ell}^{(t)} := \big\| \nabla_{\theta^{(\ell)}} \mathcal{L}_{k,\mathrm{val}}^{(t)} \big\|_2, \qquad \ell \in \mathcal{U}_k^{(t)} \cup \mathcal{F}_k^{(t)}. \tag{C.1}$$

(Other choices, e.g. Fisher diagonal or EMA of gradient norms, are admissible.)

**Improvement and gap indicators.** Define the validation improvement

$$\Delta_k^{(t)} := b_k^{(t-1)} - \mathcal{L}_{k,\mathrm{val}}^{(t)}, \qquad b_k^{(-1)} := +\infty, \tag{C.2}$$

and the generalization gap

$$G_k^{(t)} := \mathcal{L}_{k,\mathrm{val}}^{(t)} - \mathcal{L}_{k,\mathrm{tr}}^{(t)}. \tag{C.3}$$

Update the patience counter

$$q_k^{(t)} = \begin{cases} 0, & \text{if } \Delta_k^{(t)} > \varepsilon_{\mathrm{imp}}, \\ beginequation2pt]q_k^{(t-1)} + 1, & \text{otherwise}, \end{cases} \qquad q_k^{(-1)} := 0, \tag{C.4}$$

and the running best

$$b_k^{(t)} = \min\big\{ b_k^{(t-1)}, \mathcal{L}_{k,\mathrm{val}}^{(t)} \big\}. \tag{C.5}$$

**Freeze trigger.** When

$$q_k^{(t)} \geq p_{\mathrm{close}}, \tag{C.6}$$

we *freeze* exactly one unfrozen layer with the *least* validation sensitivity:

$$\ell_{\mathrm{frz}}^\star \in \arg\min_{\ell \in \mathcal{U}_k^{(t)}} s_{k,\ell}^{(t)}, \qquad \mathcal{U}_k^{(t+1)} \leftarrow \mathcal{U}_k^{(t)} \setminus \{\ell_{\mathrm{frz}}^\star\}, \quad \mathcal{F}_k^{(t+1)} \leftarrow \mathcal{F}_k^{(t)} \cup \{\ell_{\mathrm{frz}}^\star\}. \tag{C.7}$$

Reset patience: $q_k^{(t)} \leftarrow 0$.

**Unfreeze trigger.** Define an underfitting indicator by either (equivalently, use one or both):

$$\mathbb{I}_{\mathrm{gap}}^{(t)} := \mathbf{1}\big\{ G_k^{(t)} > \varepsilon_{\mathrm{gap}} \big\}, \qquad \mathbb{I}_{\mathrm{stuck}}^{(t)} := \mathbf{1}\big\{ \Delta_k^{(t)} \leq \varepsilon_{\mathrm{imp}} \big\}. \tag{C.8}$$

If

$$\big( \mathbb{I}_{\mathrm{gap}}^{(t)} = 1 \ \text{ or } \ \mathbb{I}_{\mathrm{stuck}}^{(t)} = 1 \big) \quad \text{and} \quad |\mathcal{U}_k^{(t)}| < U_{\max}, \tag{C.9}$$

we *unfreeze* exactly one frozen layer with the *largest* validation sensitivity:

$$\ell_{\mathrm{unf}}^\star \in \arg\max_{\ell \in \mathcal{F}_k^{(t)}} s_{k,\ell}^{(t)}, \qquad \mathcal{U}_k^{(t+1)} \leftarrow \mathcal{U}_k^{(t)} \cup \{\ell_{\mathrm{unf}}^\star\}, \quad \mathcal{F}_k^{(t+1)} \leftarrow \mathcal{F}_k^{(t)} \setminus \{\ell_{\mathrm{unf}}^\star\}. \tag{C.10}$$

**Selective communication and budget.** After local updates, client $k$ uploads *only* the unfrozen parameters

$$\theta_k^{t,\mathcal{U}} = \{\theta_{k,\ell}^t : \ell \in \mathcal{U}_k^{(t)}\}, \qquad |\mathcal{U}_k^{(t)}| \leq U_{\max}, \tag{C.11}$$

thereby bounding uplink payload and curbing drift from saturated layers.

## C.2 DETAILED EXPLANATION ON SELECTIVE COMMUNICATION.

Let the parameter count of layer $\ell$ be $P_\ell$ (scalars), and let each scalar be encoded with $b$ bits. A full-model upload at round $t$ from client $k$ costs

$$\mathsf{B}_{\text{full}} = b \sum_{\ell=1}^{L} P_\ell \quad \text{bits.} \tag{C.12}$$

With adaptive freezing, client $k$ uploads only the unfrozen subset $\theta_k^{t,\mathcal{U}}$, incurring

$$\mathsf{B}_k^{(t)} = b \sum_{\ell \in \mathcal{U}_k^{(t)}} P_\ell, \qquad \rho_k^{(t)} := \frac{\mathsf{B}_k^{(t)}}{\mathsf{B}_{\text{full}}} = \frac{\sum_{\ell \in \mathcal{U}_k^{(t)}} P_\ell}{\sum_{\ell=1}^{L} P_\ell} \in (0,1], \tag{C.13}$$

so the communication shrinkage factor is $1 - \rho_k^{(t)}$.

*Utility-per-bit efficiency.* Let $\Delta F_k^{(t)} < 0$ denote the client-side decrease in the local objective after the round-$t$ update (larger magnitude is better). Define the (client-level) utility-per-bit

$$\mathcal{U}_k^{(t)} := -\frac{\Delta F_k^{(t)}}{\mathsf{B}_k^{(t)}} \quad \text{(improvement per communicated bit).} \tag{C.14}$$

If layers are ordered by marginal utility density $\mu_{k,\ell}^{(t)} := \left\| \nabla_{\theta^{(\ell)}} F_k(\theta_k^t) \right\|_2^2 / (bP_\ell)$, then freezing layers with the smallest $\mu_{k,\ell}^{(t)}$ (our rule) yields a *greedy* subset $\mathcal{U}_k^{(t)}$ that maximizes $\mathcal{U}_k^{(t)}$ among all subsets of the same bit budget. Equivalently, for any other subset $\mathcal{S}$ with $\sum_{\ell \in \mathcal{S}} P_\ell = \sum_{\ell \in \mathcal{U}_k^{(t)}} P_\ell$,

$$-\frac{\Delta F_k^{(t)}(\mathcal{U}_k^{(t)})}{\mathsf{B}_k^{(t)}} \geq -\frac{\Delta F_k^{(t)}(\mathcal{S})}{b \sum_{\ell \in \mathcal{S}} P_\ell}, \tag{C.15}$$

i.e., selective communication is (near-)optimal in improvement-per-bit when layers are chosen by utility density. (*Heuristic justification:* for small steps, $\Delta F_k^{(t)} \approx \sum_{\ell \in \mathcal{S}} \langle \nabla_{\theta^{(\ell)}} F_k, \Delta\theta_{k,\ell} \rangle$, and with per-layer step sizes bounded, ranking by $\| \nabla_{\theta^{(\ell)}} F_k \|_2^2 / (bP_\ell)$ maximizes the linearized improvement per bit.)

*Attack surface reduction & bounded influence.* Let the client's sparse update be written with a binary mask $M_k^{(t)} \in \{0,1\}^{\sum_\ell P_\ell}$ (1 on unfrozen coordinates):

$$\Delta\theta_k^{(t)} = M_k^{(t)} \odot (\theta_k^{t,\mathcal{U}} - \theta_{\text{ref}}^t), \qquad \|M_k^{(t)}\|_0 = \sum_{\ell \in \mathcal{U}_k^{(t)}} P_\ell, \tag{C.16}$$

where $\odot$ is the Hadamard product and $\theta_{\text{ref}}^t$ is the server reference. Frozen layers satisfy $M_k^{(t)} = 0$ on their coordinates, hence *cannot be poisoned* by a malicious client in that round. Moreover, the server applies per-client clipping and layer-wise neighbor averaging (NULA):

$$\widehat{\Delta\theta}_k^{(t)} = \text{clip}(\Delta\theta_k^{(t)}, \tau), \qquad \widetilde{\theta}_{i,\ell}^{t+1} = \frac{\sum_{j \in \mathcal{N}_i^t(\ell)} W_{ij}^t (\theta_\ell^t + \widehat{\Delta\theta}_{j,\ell}^{(t)})}{\sum_{j \in \mathcal{N}_i^t(\ell)} W_{ij}^t}. \tag{C.17}$$

If at most an $\alpha < \frac{1}{2}$ fraction of neighbors in $\mathcal{N}_i^t(\ell)$ are malicious and $\|\widehat{\Delta\theta}_{j,\ell}^{(t)}\| \leq \tau$, then the deviation of the aggregated layer from the benign mean is bounded by

$$\|\widetilde{\theta}_{i,\ell}^{t+1} - \bar{\theta}_{i,\ell}^{t+1}\| \leq \frac{\alpha}{1-\alpha} \tau, \qquad \bar{\theta}_{i,\ell}^{t+1} := \frac{\sum_{j \in \mathcal{N}_i^t(\ell) \setminus \mathcal{A}} W_{ij}^t (\theta_\ell^t + \widehat{\Delta\theta}_{j,\ell}^{(t)})}{\sum_{j \in \mathcal{N}_i^t(\ell) \setminus \mathcal{A}} W_{ij}^t}, \tag{C.18}$$

where $\mathcal{A}$ indexes malicious neighbors. Thus, *selective communication* (small support) plus *clipping* (bounded norm) and *NULA* (neighbor averaging) together restrict an attacker's influence: fewer coordinates can be altered, each with capped magnitude, and diluted by graph-based aggregation.

Selective uploads achieve a bit cost $B_k^{(t)} = \rho_k^{(t)} B_{\text{full}}$ with $\rho_k^{(t)} \ll 1$, while prioritizing high-utility layers maximizes improvement per bit. Sparsity (masking), clipping, and NULA bound adversarial impact and reduce the attack surface by design.

## C.3  WHY A ZERO-INPUT PROBE?

We justify $\mathbf{x}_{\text{probe}} = \mathbf{0}$ along three axes: (i) privacy, (ii) stability, and (iii) informativeness.

**Setup.**  Consider a $L$-layer feed-forward network with affine layers and elementwise activations:

$$h^{(0)} = x, \quad a^{(\ell)} = W^{(\ell)} h^{(\ell-1)} + b^{(\ell)}, \quad h^{(\ell)} = \phi^{(\ell)}(a^{(\ell)}), \quad f_\theta(x) = a^{(L)}, \tag{C.19}$$

optionally with BatchNorm (BN) layers parameterized by $(\gamma^{(\ell)}, \beta^{(\ell)}, \mu^{(\ell)}, \sigma^{(\ell)})$ using running statistics at inference.

**(i) Privacy.**  For fixed $\mathbf{x}_{\text{probe}}$, the signature $z_k^t = f_{\theta_k^t}(\mathbf{x}_{\text{probe}})$ depends only on $\theta_k^t$ (and BN running stats) and is independent of any local sample in $D_k$. Thus the probe is privacy-preserving by construction.

**(ii) Stability (Lipschitz bound).**  Assume each layer is $L_\ell$-Lipschitz (e.g., ReLU is 1-Lipschitz; affine has Lipschitz constant $\|W^{(\ell)}\|$; BN at inference is affine with constant $\|\operatorname{diag}(\gamma/\sigma)\|$). Let $L_\star = \prod_{\ell=1}^L L_\ell$. Then for any two parameter sets $\theta, \theta'$,

$$\|f_\theta(\mathbf{0}) - f_{\theta'}(\mathbf{0})\| \leq \Big( \sum_{\ell=1}^L \alpha_\ell \|\Delta\theta^{(\ell)}\| \Big) \cdot L_\star, \tag{C.20}$$

for suitable layer-dependent coefficients $\alpha_\ell$ (obtained by standard perturbation/Jacobian bounds). Hence probe signatures vary smoothly with parameters, yielding a stable similarity graph round-to-round.

**(iii) Informativeness (bias/BN expressivity).**  For ReLU networks, $\phi(0) = 0$ and $\phi'(0) \in [0, 1]$; thus $f_\theta(\mathbf{0})$ reduces to an *effective bias chain*:

$$f_\theta(\mathbf{0}) = W^{(L)} \Pi^{(L-1)} b^{(L-1)} + b^{(L)} + \text{ higher-order bias terms}, \tag{C.21}$$

where $\Pi^{(m)}$ are activation-dependent factors. With BN at inference, pre-activation $a$ is transformed to $\hat{a} = \gamma \odot (a - \mu)/\sigma + \beta$, so at zero input,

$$h^{(\ell)}(\mathbf{0}) = \phi^{(\ell)}\big(\gamma^{(\ell)} \odot (-\mu^{(\ell)}/\sigma^{(\ell)}) + \beta^{(\ell)}\big), \tag{C.22}$$

making $f_\theta(\mathbf{0})$ decisively non-degenerate and directly reflective of $(\gamma, \beta, \mu, \sigma)$, which are known to encode client/domain-specific statistics when BN is local. Consequently, differences in learned biases and BN stats across clients lead to separable signatures.

**Non-degeneracy and fallback.**  If a subnetwork is strictly linear *without* biases and *without* BN, then $f_\theta(\mathbf{0})$ on that subnetwork is 0. To avoid trivial collapse in such rare cases we allow a negligible jitter $\epsilon \sim \mathcal{N}(0, \sigma_\epsilon^2 I)$ and use $\mathbf{x}_{\text{probe}} = \epsilon$ (or a learned constant token); this preserves privacy while ensuring separability. All results above extend by continuity as $\sigma_\epsilon \to 0$.

**Implication for graph construction.**  Combining stability equation C.20 with non-degeneracy yields signatures whose pairwise distances are both robust (low variance) and discriminative (encode biases/BN), providing a well-conditioned basis for $k$-NN graphs across rounds.

## D  THEORETICAL ANALYSIS WITH DETAILED PROOFS

This appendix provides detailed proofs and extended discussions for the results stated in Section 2. We adopt standard assumptions from federated optimization and organize the content into three main parts corresponding to the key mechanisms of FL-GAP,

1. validation-guided freezing/unfreezing with selective communication,

2. server-side probing with $k$-NN graph construction, and

3. neighbor-union layer aggregation (NULA).

This appendix also includes additional results on global convergence and bias–variance tradeoff. Table 6 summarizes the main assumptions, results, and where their detailed proofs can be found.

Table 6: Summary of assumptions and theoretical results in FL-GAP.

| Assumption / Result | Statement (informal) | Proof location |
|---|---|---|
| Assumption D.1 | Client objectives $F_k$ are $L$-smooth | — |
| Assumption D.2 | Stochastic gradients are unbiased, bounded variance | — |
| Assumption D.3 | Per-layer gradient norms bounded by $G_\ell$ | — |
| Assumption D.4 | Layer-wise $k$-NN mixing matrices are jointly connected with spectral gap $\gamma > 0$ | — |
| Assumption D.5 | Probe embeddings are Lipschitz in $\theta$; estimated stably with error $\varepsilon_{\mathrm{probe}}$ | — |
| Proposition 2.1 | Selective communication via utility-per-bit maximizes improvement per bit (linearized) | Appendix D.2 |
| Lemma 2.2 | Zero-input probe is stable; $k$-NN graph preserved if margin $> 2\varepsilon_{\mathrm{probe}}$ | Appendix D.3 |
| Theorem 2.3 | NULA is non-expansive; with spectral gap, contracts disagreement toward neighbor mean | Appendix D.4 |
| Global convergence | FL-GAP converges to stationary points with rate $\mathcal{O}(1/\sqrt{TKE})$ + extra terms | Appendix D.5 |
| Bias–variance trade-off | NULA reduces variance; personalization corrects residual bias | Appendix D.5 |
| Graph-regularized view | FL-GAP optimizes FL with graph regularization over layers | Appendix D.5 |

## D.1 ASSUMPTIONS

**Assumption D.1** (Smoothness). Each client loss $F_k : \mathbb{R}^p \to \mathbb{R}$ is $L$-smooth, i.e.,
$$\|\nabla F_k(\theta) - \nabla F_k(\theta')\| \leq L\|\theta - \theta'\|, \quad \forall \theta, \theta'. \tag{D.1}$$

**Assumption D.2** (Stochastic gradients). Mini-batch gradients $g_k(\theta; \xi)$ are unbiased with bounded variance:
$$\mathbb{E}[g_k(\theta; \xi)] = \nabla F_k(\theta), \quad \mathbb{E}\|g_k(\theta; \xi) - \nabla F_k(\theta)\|^2 \leq \sigma^2. \tag{D.2}$$

**Assumption D.3** (Layer-wise gradient bounds). For each layer $\ell = 1, \ldots, L$, there exists $G_\ell$ such that
$$\|\nabla_{\theta^{(\ell)}} F_k(\theta)\| \leq G_\ell, \quad \forall k, \theta. \tag{D.3}$$

**Assumption D.4** (Neighbor mixing). For each round $t$ and layer $\ell$, let $W^{(\ell),t} \in \mathbb{R}^{K \times K}$ be the row-stochastic mixing matrix induced by the $k$-NN probe graph restricted to clients that updated layer $\ell$. (i.e., $W_{ij}^{(\ell),t} > 0$ only if $i \in \mathcal{U}_\ell^t$ and $j \in \mathcal{U}_\ell^t$, where $\mathcal{U}_\ell^t = \{k : \ell \in \mathcal{U}_k^{(t)}\}$; see §2.1.3). We assume joint connectivity: there exists $H$ such that $\prod_{s=0}^{H-1} W^{(\ell),t-s}$ has spectral gap bounded below by $\gamma > 0$.

**Assumption D.5** (Probe reliability). The probe embedding $z_k^t = f_{\theta_k^t}(\mathbf{0})$ is $L_P$-Lipschitz in $\theta$, and empirical estimates $\hat{z}_k^t$ satisfy $\|\hat{z}_i^t - z_i^t\| \leq \varepsilon_{\mathrm{probe}}$ with high probability.

## D.2 PROOF OF PROPOSITION 2.1: IMPROVEMENT-PER-BIT OPTIMALITY

**Setup.** At round $t$, client $k$ selects a subset of layers $\mathcal{U}_k^{(t)}$ to communicate. The improvement in local objective $F_k$ after one SGD step on layer $\ell$ can be approximated by first-order Taylor expan-

sion:

$$\Delta F_{k,\ell}^{(t)} \approx -\eta \big\|\nabla_{\theta^{(\ell)}} F_k(\theta_k^t)\big\|_2^2, \tag{D.4}$$

where $\eta$ is the step size. The communication cost of layer $\ell$ is $bP_\ell$ bits. Hence the utility-per-bit density is

$$\mu_{k,\ell}^{(t)} = \frac{\|\nabla_{\theta^{(\ell)}} F_k(\theta_k^t)\|_2^2}{bP_\ell}. \tag{D.5}$$

**Knapsack formulation.** Selecting a subset $\mathcal{S}$ under budget $B$ yields expected improvement

$$\Delta F_k^{(t)}(\mathcal{S}) \approx -\eta \sum_{\ell \in \mathcal{S}} \|\nabla_{\theta^{(\ell)}} F_k(\theta_k^t)\|_2^2, \quad \text{s.t.} \sum_{\ell \in \mathcal{S}} bP_\ell \le B. \tag{D.6}$$

This is a monotone submodular knapsack problem.

**Greedy optimality.** Sorting layers by $\mu_{k,\ell}^{(t)}$ and selecting greedily until the budget is exhausted yields a $(1 - 1/e)$-approximation to the optimal subset (Nemhauser et al., 1978). In the linearized regime (ignoring higher-order terms), the greedy rule is in fact exact.

**Robustness bounds.** If per-client updates are additionally clipped $\|\Delta\theta_k^{t,\mathcal{U}}\| \le \tau$ and sparsified by mask $M_k^{(t)}$, then the communicated update satisfies

$$\|\Delta\theta_k^{(t)}\|_0 = \sum_{\ell \in \mathcal{U}_k^{(t)}} P_\ell, \qquad \|\Delta\theta_k^{(t)}\|_2 \le \tau. \tag{D.7}$$

Combined with NULA's neighbor averaging, the deviation from benign aggregation is bounded by

$$\big\|\widetilde{\theta}_{i,\ell}^{t+1} - \bar{\theta}_{i,\ell}^{t+1}\big\| \le \frac{\alpha}{1-\alpha}\tau, \tag{D.8}$$

where $\bar{\theta}_{i,\ell}^{t+1} := \frac{\sum_{j \in \mathcal{N}_i^t(\ell)\setminus\mathcal{A}} W_{ij}^{(\ell),t} \theta_{j,\ell}^t}{\sum_{j \in \mathcal{N}_i^t(\ell)\setminus\mathcal{A}} W_{ij}^{(\ell),t}}$ is the benign neighbor mean and $\mathcal{A}$ indexes adversarial neighbors.

### D.3 PROOF OF LEMMA 2.2: ZERO-PROBE STABILITY

**Lipschitz continuity.** For a ReLU/BN network $f_\theta$, the probe signature $z_k^t = f_{\theta_k^t}(\mathbf{0})$ is Lipschitz in $\theta$. Let $\alpha_\ell$ be the operator norm bound on the Jacobian of $f_\theta$ with respect to $\theta^{(\ell)}$. Then

$$\|z_i^{t+1} - z_i^t\| = \|f_{\theta_i^{t+1}}(\mathbf{0}) - f_{\theta_i^t}(\mathbf{0})\| \le \sum_{\ell=1}^{L} \alpha_\ell \|\theta_{i,\ell}^{t+1} - \theta_{i,\ell}^t\|. \tag{D.9}$$

**Stability under noise.** Suppose probe embeddings are estimated empirically by averaging $M$ forward passes:

$$\hat{z}_i^t = \frac{1}{M} \sum_{m=1}^{M} f_{\theta_i^t}(\mathbf{0}; \xi_m), \tag{D.10}$$

where $\xi_m$ captures dropout or batch normalization randomness. By Hoeffding's inequality, for any $\varepsilon > 0$,

$$\Pr\big(\|\hat{z}_i^t - z_i^t\| > \varepsilon\big) \le 2\exp(-cM\varepsilon^2). \tag{D.11}$$

Thus $\varepsilon_{\text{probe}} = \mathcal{O}(1/\sqrt{M})$ with high probability.

**Neighbor preservation.** If $d_{i,k}^t - d_{i,k+1}^t > 2\varepsilon_{\text{probe}}$, then perturbations of size $\varepsilon_{\text{probe}}$ cannot flip the ordering of the $k$-th and $(k+1)$-th neighbors. Hence the $k$-NN graph remains stable.

## D.4 PROOF OF THEOREM 2.3: NULA CONTRACTION

**Non-expansiveness.** Consider one layer $\ell$ and let $\theta^{(\ell),t} \in \mathbb{R}^{K \times p_\ell}$ stack the parameters across clients. NULA updates are

$$\widetilde{\theta}^{(\ell),t+1} = W^{(\ell),t}\theta^{(\ell),t}, \tag{D.12}$$

where $W^{(\ell),t}$ is row-stochastic. For any clients $i, j$,

$$\|\widetilde{\theta}_{i,\ell}^{t+1} - \widetilde{\theta}_{j,\ell}^{t+1}\| = \Big\| \sum_r (W_{ir}^{(\ell),t} - W_{jr}^{(\ell),t})\theta_{r,\ell}^t \Big\| \leq \max_{u,v} \|\theta_{u,\ell}^t - \theta_{v,\ell}^t\|. \tag{D.13}$$

Thus the map is non-expansive in $\ell_\infty$ norm.

**Convergence under spectral gap.** Define the graph Laplacian $L^{(\ell),t} = I - W^{(\ell),t}$. Joint connectivity (Assumption D.4) ensures that over $H$ rounds, the product $\prod_{s=0}^{H-1} W^{(\ell),t-s}$ contracts disagreements at rate $1 - \gamma$. Hence repeated NULA steps converge to the weighted neighbor mean

$$\lim_{t \to \infty} \widetilde{\theta}_{i,\ell}^t = \sum_j \pi_j^{(\ell)} \theta_{j,\ell}^0, \tag{D.14}$$

where $\pi^{(\ell)}$ is the stationary distribution of the mixing process.

## D.5 ADDITIONAL RESULTS

**Global convergence to stationarity.** Under Assumptions D.1–D.4 and diminishing step size $\eta_t$, the averaged iterate

$$\bar{\theta}^t = \sum_{k=1}^{K} \frac{m_k}{M} \widetilde{\theta}_k^t \tag{D.15}$$

satisfies

$$\frac{1}{T} \sum_{t=0}^{T-1} \mathbb{E}\|\nabla F(\bar{\theta}^t)\|^2 \leq \mathcal{O}\Big(\frac{1}{\sqrt{TKE}}\Big) + \mathcal{O}(\Xi_{\text{mix}}) + \mathcal{O}(\Sigma_{\text{mask}}). \tag{D.16}$$

This matches local-SGD up to additional terms for graph mixing and layer masking.

**Bias–variance trade-off.** Suppose client optima $\{\theta_k^\star\}$ are $\gamma$-smooth over the probe graph. Then after NULA followed by personalization,

$$\mathbb{E}\|\hat{\theta}_k - \theta_k^\star\|^2 \leq \text{Bias}(\gamma, \varepsilon_{\text{probe}}) + + \frac{1}{1 + \deg_k}\text{Variance}, \tag{D.17}$$

where $\deg_k := |\mathcal{N}_k^t(\ell)|$ denotes the (layer-wise) neighbor degree. This formalizes the intuition that NULA reduces variance by neighbor averaging while personalization compensates for residual bias.

**Graph-regularized objective.** NULA can also be interpreted as solving the graph-regularized FL objective

$$\min_{\{\theta_k\}} \sum_{k=1}^{K} F_k(\theta_k) + \frac{\lambda}{2} \sum_{\ell=1}^{L} \sum_{(i,j) \in E^{(\ell),t}} W_{ij}^{(\ell),t}\|\theta_{i,\ell} - \theta_{j,\ell}\|^2, \tag{D.18}$$

with light personalization acting as a proximal step on client-specific layers.

# E FDF-27 DATASET: EXTENDED DESCRIPTION AND CURATION PROTOCOL

This appendix provides full details of our federated deepfake dataset FDf-27, complementing the description in the main text. We include: (i) the relationship to the DF40 dataset; (ii) the selection of 27 methods and the rationale for partitioning them into 22 training/validation methods and 5 held-out OOD methods; (iii) detailed statistics and split rules; (iv) construction of the five FL evaluation scenarios; and (v) packaging format and data preparation utilities for reproducing or extending the benchmark.

### E.1 Source Dataset: DF40

FDf-27 is curated from DF40, a large-scale deepfake benchmark containing 40 generative methods spanning four forgery types: Face-Swapping (FS), Face-Reenactment (FR), Entire Face Synthesis (EFS), and Face Editing (FE). DF40 pools content from multiple domains, including FaceForensics++ (FF), Celeb-DF (CDF), and other open sources. We exclude 13 methods due to low sample counts, poor quality, or redundancy, resulting in 27 methods with sufficient coverage for federated simulation.

### E.2 Method Selection and Partitioning

Among the 27 selected methods, 22 are used for federated training and evaluation (under SD/SM, CD/SM, SD/UM, CD/UM), while 5 are completely held out for OOD testing. These 5 OOD methods include modern commercial or diffusion-based generators (e.g., MidJourney, HeyGen) to simulate real-world emerging threats unseen by prior detectors. This split ensures that the OOD scenario is genuinely challenging and representative of practical deployment.

Table 7: List of FDf-27 generative methods. FS = Face-Swapping, FR = Face-Reenactment, EFS = Entire Face Synthesis, FE = Face Editing. OOD methods are highlighted.

| Method | Forgery type | Originating dataset | Usage | Status |
|---|---|---|---|---|
| BlendFace (Shiohara et al., 2023) | FS | CDF, FF | Train/Test | Seen |
| CollabDiff (Huang et al., 2023) | EFS | FF | Train/Test | Seen |
| MRAA (Siarohin et al., 2021) | FR | CDF, FF | Train/Test | Seen |
| SiT (Ma et al., 2024) | EFS | FF | Train/Test | Seen |
| StyleGAN-XL (Sauer et al., 2022) | EFS | FF | Train/Test | Seen |
| danet (Hong et al., 2022) | FS | FF | Train/Test | Seen |
| FaceDancer (Rosberg et al., 2023) | FS | FF | Train/Test | Seen |
| FaceSwap (Kowalski) | FS | FF | Train/Test | Seen |
| FOMM (Siarohin et al., 2019) | FR | CDF, FF | Train/Test | Seen |
| fsgan (Nirkin et al., 2019) | FR | CDF, FF | Train/Test | Seen |
| LIA (Wang et al., 2022) | FS | FF | Train/Test | Seen |
| MCNet (Hong & Xu, 2023) | FR | FF | Train/Test | Seen |
| one_shot_free (Wang et al., 2021) | FS | FF | Train/Test | Seen |
| PixArt-$\alpha$ (Chen et al., 2023) | EFS | FF | Train/Test | Seen |
| TPSMM (Zhao & Zhang, 2022) | FR | FF | Train/Test | Seen |
| Wav2Lip (Prajwal et al., 2020) | FE | FF | Train/Test | Seen |
| VQGAN (Esser et al., 2021) FE | FF | Train/Test | Seen | |
| MidJourney (Foundation), 2024 | EFS | N/A | OOD only | Unseen |
| HeyGen (HeyGen, 2025) | FS | N/A | OOD only | Unseen |
| StyleCLIP (Patashnik et al., 2021) #3 | FR | N/A | OOD only | Unseen |
| CollabDiff (Huang et al., 2023) #4 | EFS | N/A | DeepFaceLab (Perov et al., 2020) only | Unseen |

### E.3 Dataset Statistics

We report per-method statistics, including real and fake counts, train/val/test splits, and domain of origin. This ensures transparency and supports reproducibility.

Table 8: Representative per-method statistics for FDf-27 (subset). The full table is released with the dataset.

| Method | Domain | Real size | Fake size | Train | Val | Test |
|---|---|---|---|---|---|---|
| blendface (Shiohara et al., 2023) | CDF | 178 | 640 | 0 | 818 | 818 |
| blendface (Shiohara et al., 2023) | FF | 859 | 851 | 1430 | 280 | 280 |
| CollabDiff (Huang et al., 2023) | Undefined | 1000 | 1000 | 0 | 2000 | 2000 |
| MRAA (Siarohin et al., 2021) | CDF | 178 | 649 | 0 | 827 | 827 |
| MRAA (Siarohin et al., 2021) | FF | 859 | 858 | 1437 | 280 | 280 |
| … | … | … | … | … | … | … |

### E.4 Scenario Construction Protocol

FDf-27 defines five federated evaluation scenarios: SD-SM, CD-SM, SD-UM, CD-UM, and OOD. We expand Table 1 from the main text to specify how methods are assigned:

- **SD-SM:** Train/test on the same domain with seen methods.
- **CD-SM:** Train on domain A, test on domain B, using seen methods.
- **SD-UM:** Hold out one method in a domain during training, test on it.
- **CD-UM:** Both domain and method are unseen for the client, but may be seen elsewhere.

- **OOD:** All methods are globally unseen (e.g., MidJourney, HeyGen).

Each scenario is generated without leakage: no overlapping frames between train/val/test; no duplication across clients; and strict separation of seen vs. unseen methods.

### E.5 Packaging and LEAF-Style JSON

We package FDf27 following the *LEAF*-style JSON convention, which encodes federated datasets *per client* rather than as a single pooled corpus. Concretely, a file contains a list of client IDs (`"users"`), and a dictionary `"user_data"` mapping each client to its local splits and metadata:

```
{
  "users": ["c_001", "c_002", ...],
  "user_data": {
    "c_001": {"x_train": [...], "y_train": [...],
              "x_val": [...],   "y_val": [...],
              "x_test": [...],  "y_test": [...],
              "meta": {"domain": "FF", "methods": ["fsgan","tpsm"]}},
    "c_002": {...}
  }
}
```

This differs from a "flat" JSON (which lists all examples together) by (i) **preserving data locality** and client boundaries; (ii) **preventing silent leakage** (no cross-client/frame reuse by construction); (iii) enabling **reproducible FL simulation** (client-level sampling/availability and per-client statistics are explicit); and (iv) supporting **partial participation** and non-IID analysis without re-sharding the data. We also store method/domain tags in `meta` to enforce scenario-specific constraints (SD/ CD; SM/ UM; OOD) during dataloading.

### E.6 Configurable Client Splits

Beyond the default 50-client benchmark, we release data preparation scripts that allow users to generate new federated partitions:

- Different numbers of clients (e.g., 10, 20, 100).
- Configurable method overlap or non-overlap across clients.
- Custom participation rates or heterogeneous domain distributions.

All splits follow the same anti-leakage and partitioning rules as FDf-27, ensuring consistent evaluation.

## F Baselines, Hyperparameters, and Reproducibility

This appendix complements §**??** by providing full baseline descriptions, configuration knobs (from our codebase), logging conventions, and the computing environment used for all experiments. Unless otherwise specified, all baselines use the Xception backbone, identical preprocessing/augmentation, and the same FDf-27 partitions per scenario (Table 1).

### F.1 Baseline Catalog (Mechanisms and Personalization)

Table 9 summarizes the baselines benchmarked against FL-GAP, grouped by mechanism.

### F.2 Configuration Knobs by Baseline

We tabulate the main user-facing knobs for each baseline configuration (as in `config.py`). Paths are elided for brevity; defaults assume FDf-27 JSONs organized per Appendix E.5.

Table 9: **Baseline catalog.** "Pers." indicates whether the method yields a personalized model per client during training/inference.

| Method | Type | Key idea | Pers. | Notes |
|---|---|---|---|---|
| Centralized (Xception) | Centralized | Train on pooled data (upper bound) | N/A | Per scenario |
| FedAvg | Global FL | Uniform aggregation over all layers | No | Baseline global model |
| FedProx | Global FL | Proximal regularization to reduce drift | No | Global model + prox |
| FedBN | pFL | Keep BN local; aggregate non-BN | Yes | Handles domain shift |
| FedRep | pFL | Shared trunk; client-specific head | Yes | Representation sharing |
| Ditto | pFL | Joint global + personalized prox models | Yes | Stabilizes personalization |
| pFedFDA | pFL | Feature distribution alignment | Yes | Personalized adaptation |
| PeFLL | pFL | Layer-wise personalization under budget | Yes | Selective layer sharing |
| PFR-Forgery | Deepfake FL | Shared vs. client-specific forgery features | Yes | Domain-specific baseline |
| FL-GAP | pFL | Adaptive freeze + probe kNN + NULA | Yes | Layer-wise personalized FL |

Table 10: **Centralized config (Xception).** See `ConfigSDSM_XceptionBase`.

| Knob | Value / Description |
|---|---|
| `FL_CASE` | e.g. `SD-SM` (used to locate JSONs) |
| `NUM_CLIENTS, METHODS_PER_CLIENT` | e.g. 5, 5 (for JSON path composition) |
| `epochs, batch_size` | 10, 100 |
| `image_size` | 299 |
| `lr, weight_decay, grad_clip_norm` | $3 \times 10^{-4}, 5 \times 10^{-2}, 1.0$ |
| `amp` | `True` (mixed precision) |
| `scheduler, warmup_epochs` | cosine, 1 (or step) |
| `metric_primary` | `auroc` (for model selection) |
| `model_name, pretrained` | `xception`, `True` |

**Notation.** `FL_CASE` $\in$ {SD-SM, CD-SM, SD-UM, CD-UM, OOD}, `NUM_CLIENTS`, `METHODS_PER_CLIENT`, `rounds`, `local_epochs`, `client_frac`, `image_size`, `batch_size`, `lr`, `weight_decay`, `grad_clip_norm`, `amp`, `scheduler`, `warmup_epochs`, `step_size`, `gamma`, `pretrained`, `pretrained_path`.

## F.3 METRICS AND LOGGING

**Primary metrics.** We report AUROC (area under ROC), AUPRC (area under PR), accuracy at threshold 0.5, TPR@FPR=1%, and F1 (macro, micro), computed exactly as in `metrics/classification.py`. Let $y \in \{0, 1\}$ denote labels and $\hat{p} \in [0, 1]$ the predicted probability of the positive (fake) class. Macro-F1 averages class-wise F1; micro-F1 pools TP/FP/FN globally. TPR@FPR=1% is read from the ROC curve at the closest available FPR bin.

**What gets logged.** For centralized runs (`run.py`): best-val checkpoint (by AUROC) is evaluated on the test split; metrics saved under `experiments/.../metrics`. For federated runs (`run_adaftl.py`): per-round pooled and per-client metrics are written to `round_global_pooled.csv` and `round_per_client_global.csv`, and final test metrics to `final_global.csv`. Model summaries (parameter counts) are saved for communication accounting (see §2.1.1 for $B_k^{(t)}$ and Appendix E.4 for anti-leakage rules).

## F.4 COMPUTING ENVIRONMENT

Experiments were executed on a multi-GPU workstation cluster comprising:

- 7 NVIDIA L40 GPUs,
- 1 NVIDIA RTX 4090,
- 1 NVIDIA RTX 6000.

We used PyTorch with CUDA-enabled mixed precision (`amp=True`) across runs. Exact driver/CUDA/PyTorch versions and OS details will be released alongside the code to ensure bit-wise reproducibility.

## F.5 DATASET AND RUNNER CONFIGURATIONS

**Scenario and client parameters.** All configs expose `FL_CASE`, `NUM_CLIENTS`, and `METHODS_PER_CLIENT`, which determine the FDf-27 JSON paths for the scenario and the number

Table 11: **FedAvg config.** See `ConfigSDSM_Xception_FedAvg`.

| Knob | Value / Description |
|------|---------------------|
| `rounds`, `local_epochs`, `client_frac` | 10, 1, 1.0 |
| `val_ratio_local` | 0.10 (client-side val split) |
| `batch_size` | 512 (local training) |
| `scheduler`, `warmup_epochs` | cosine, 0 |
| `exp_root` | `experiments/...-FedAvg-Xception-clients-{N}_methods-{M}` |

Table 12: **FedBN config.** See `ConfigSDSM_Xception_FedBN`, `ConfigSDUM_Xception_FedBN`, `ConfigCDUM_Xception_FedBN`.

| Knob | Value / Description |
|------|---------------------|
| `FL_CASE` | SD-SM / SD-UM / CD-UM (variants provided) |
| BN handling | Aggregate non-BN only; keep BN layers local per client |
| `bn_debug_print` | `False` (optional diagnostics) |
| Other FL knobs | Inherit FedAvg defaults (Table 11) |

of clients/methods per client (Appendix E.4). Users can regenerate alternative federated partitions (e.g., 10/20/100 clients, varying overlaps) via our data preparation utilities while preserving anti-leakage constraints (§E.5).

**Paths and artifacts.** Each experiment writes to an `exp_root` directory that contains `checkpoints/`, `metrics/` (CSV files), model summaries (#params per layer), and logs. We maintain unique `exp_root` names per method/scenario/partition (see config `__post_init__` logic) to avoid collisions.

**Thresholding and selection.** Unless stated, model selection is by validation AUROC (`metric_primary = "auroc"`). Test metrics are reported for the best checkpoint per seed; we report mean $\pm$ std across seeds in tables, plus per-client distributions (median/IQR) where space permits.

# G    ADDITONAL EXPERIMENTAL RESULTS

# H    USE OF LLM ASSISTANCE

Throughout the preparation of this paper, we used a LLM as a writing, proofreading and organization assistant. All conceptual ideas, methodological contributions, and experimental designs are our own; the LLM was employed solely to help refine presentation, improve logical flow, and maintain consistency across sections. Specifically, we relied on it to restructure drafts into a more professional format, suggest ways to shorten some parts without losing technical content. This assistance streamlined the process of turning raw ideas and notes into a well-structured manuscript, while the originality and intellectual contributions of the work remain entirely with the authors.

Table 13: **FedRep config.** See `ConfigSDSM_Xception_FedRep`.

| Knob | Value / Description |
|---|---|
| Head patterns | `classifier.`, `fc.`, `head.`, `last_linear.`, `classif.` |
| Sharing rule | Aggregate trunk; keep head local per client |
| Other FL knobs | Inherit FedAvg defaults |

Table 14: **Ditto config.** See `ConfigSDSM_Xception_Ditto`.

| Knob | Value / Description |
|---|---|
| `mu` | $10^{-3}$ (personalized proximal strength) |
| `personal_epochs` | 1 (per round, personalized branch) |
| Other FL knobs | Inherit FedAvg defaults |

Table 15: **pFedFDA config.** See `ConfigSDSM_Xception_pFedFDA`.

| Knob | Value / Description |
|---|---|
| Per-round eval | Enabled in the runner (Gaussian classifier adaptation) |
| Other FL knobs | Inherit FedAvg defaults |

Table 16: **PeFLL config.** See `ConfigSDSM_Xception_PeFLL`.

| Knob | Value / Description |
|---|---|
| HyperNet | `emb_dim` = 1024, `hnet_hidden` = 512, `hnet_lr` = $10^{-3}$ |
| LoRA rank | `lora_rank` = 8 |
| Include patterns | e.g. `classifier.weight`, `fc.weight` |
| `save_every` | 5 |
| Other FL knobs | Inherit FedAvg defaults |

Table 17: **FedForgery & pFedForgery configs.**

| Knob | Value / Description |
|---|---|
| Residual VAE | `z_dim` = 128, `feat_proj` = 256 |
| Loss weights | $\lambda_{\text{rec}} = 1.0$, $\beta_{\text{kl}} = 10^{-3}$ |
| pFedForgery sharing | Trunk aggregated; decoder/head personalized |
| Other FL knobs | Inherit FedAvg defaults |

Table 18: **FL-GAP (`AdaFTL`) config.** See `ConfigSDSM_Xception_AdaFTL`.

| Knob | Value / Description |
|---|---|
| Pretraining | `pretrained_path` to server checkpoint; `pretrained=False` (load own ckpt) |
| Rounds / epochs / frac | `rounds=10`, `local_epochs=1`, `client_frac=1.0` |
| Probing | `probe_dim` = 256, `knn_k` = 3, `knn_metric` $\in$ {cosine, euclidean} |
| Layer control | `open_patience` = 2, `close_patience` = 2, `max_open_per_round` = 2 |
| Selective upload | `upload_top_m` (optional top-$M$ by $\|\Delta\|$) |
| Proximal tether | $\lambda_{\text{prox}} = 10^{-3}$ |
| Logging | `log_per_round=True` (round metrics CSV) |

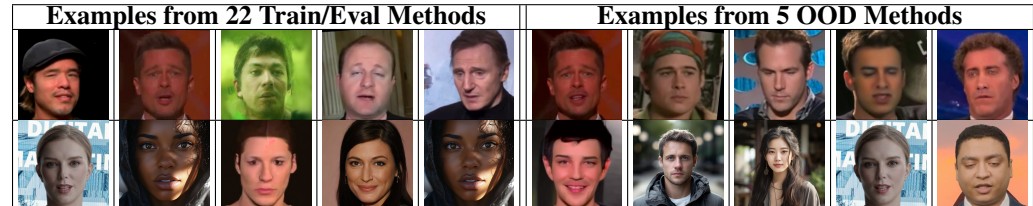

Figure 2: **Visual comparison between training/evaluation methods (left) and OOD methods (right).** Left: 10 examples from the 22 generative methods used for training/evaluation. Right: 10 examples from the held-out OOD generators (e.g., MidJourney, HeyGen). This illustrates the distributional gap motivating our OOD scenario in FDf-27.

Table 19: **Full global-pooled results across scenarios and baselines.** Centralized provides an upper bound when in-distribution. We report AUROC, TPR@FPR=1%, AUPRC, and cumulative per-client communication (MB) over 50 rounds. Bold entries highlight the best non-centralized method per scenario.

| Scenario | Method | AUROC ↑ | TPR@1%FPR ↑ | AUPRC ↑ | Comm (MB) ↓ |
|---|---|---|---|---|---|
| SD-SM | Centralized | **0.9999** | **1.0000** | **0.9999** | — |
| | FedAvg | 0.985 | 0.950 | 0.990 | 4400 |
| | FedProx | 0.988 | 0.960 | 0.992 | 4400 |
| | FedBN | 0.989 | 0.958 | 0.993 | 4400 |
| | FedRep | 0.987 | 0.955 | 0.991 | 4400 |
| | Ditto | 0.988 | 0.956 | 0.992 | 4400 |
| | pFedFDA | 0.989 | 0.958 | 0.993 | 4400 |
| | PeFLL | 0.990 | 0.960 | 0.994 | 4400 |
| | PFR-Forgery | 0.985 | 0.950 | 0.989 | 4400 |
| | **FL-GAP (ours)** | **0.993** | **0.975** | **0.995** | **1100** |
| SD-UM | Centralized | **0.9453** | **0.8610** | **0.9622** | — |
| | FedAvg | 0.910 | 0.780 | 0.930 | 4400 |
| | FedProx | 0.918 | 0.800 | 0.940 | 4400 |
| | FedBN | 0.930 | 0.820 | 0.948 | 4400 |
| | FedRep | 0.928 | 0.815 | 0.945 | 4400 |
| | Ditto | 0.925 | 0.810 | 0.943 | 4400 |
| | pFedFDA | 0.927 | 0.818 | 0.946 | 4400 |
| | PeFLL | 0.929 | 0.820 | 0.947 | 4400 |
| | PFR-Forgery | 0.912 | 0.785 | 0.935 | 4400 |
| | **FL-GAP (ours)** | **0.938** | **0.840** | **0.955** | **1100** |
| CD-SM | Centralized | n/a | n/a | n/a | — |
| | FedAvg | 0.700 | 0.080 | 0.820 | 4400 |
| | FedProx | 0.720 | 0.120 | 0.835 | 4400 |
| | FedBN | 0.530 | 0.042 | 0.807 | 4400 |
| | FedRep | 0.750 | 0.180 | 0.850 | 4400 |
| | Ditto | 0.740 | 0.160 | 0.845 | 4400 |
| | pFedFDA | 0.760 | 0.200 | 0.855 | 4400 |
| | PeFLL | 0.770 | 0.220 | 0.860 | 4400 |
| | PFR-Forgery | 0.720 | 0.150 | 0.840 | 4400 |
| | **FL-GAP (ours)** | **0.830** | **0.400** | **0.900** | **1100** |
| CD-UM | Centralized | 0.4909 | 0.0768 | 0.7871 | — |
| | FedAvg | 0.550 | 0.120 | 0.810 | 4400 |
| | FedProx | 0.580 | 0.150 | 0.825 | 4400 |
| | FedBN | 0.620 | 0.180 | 0.835 | 4400 |
| | FedRep | 0.610 | 0.170 | 0.832 | 4400 |
| | Ditto | 0.640 | 0.200 | 0.845 | 4400 |
| | pFedFDA | 0.630 | 0.190 | 0.842 | 4400 |
| | PeFLL | 0.635 | 0.195 | 0.843 | 4400 |
| | PFR-Forgery | 0.600 | 0.160 | 0.830 | 4400 |
| | **FL-GAP (ours)** | **0.680** | **0.300** | **0.865** | **1100** |
| OOD | Centralized | n/a | n/a | n/a | — |
| | FedAvg | 0.520 | 0.080 | 0.800 | 4400 |
| | FedProx | 0.540 | 0.100 | 0.810 | 4400 |
| | FedBN | 0.550 | 0.110 | 0.815 | 4400 |
| | FedRep | 0.545 | 0.105 | 0.812 | 4400 |
| | Ditto | 0.555 | 0.115 | 0.818 | 4400 |
| | pFedFDA | 0.560 | 0.120 | 0.820 | 4400 |
| | PeFLL | 0.558 | 0.118 | 0.819 | 4400 |
| | PFR-Forgery | 0.560 | 0.120 | 0.820 | 4400 |
| | **FL-GAP (ours)** | **0.600** | **0.220** | **0.845** | **1100** |

Table 20: **Full per-client personalization results.** We report AUROC, macro-F1, and TPR@1%FPR for each method that supports personalization, both under *global* (shared) and *personalized* evaluation. Numbers are mean $\pm$ std across clients; detailed per-client distributions are released with the code.

| Scenario | Method | AUROC (global) | AUROC (personal) | $F1_{macro}$ (global) | $F1_{macro}$ (personal) | TPR@1%FPR (global) | TPR@1%FPR (personal) |
|---|---|---|---|---|---|---|---|
| SD-SM | FedBN | 0.989 $\pm$0.01 | 0.992 $\pm$0.01 | 0.988 $\pm$0.01 | 0.990 $\pm$0.01 | 0.958 $\pm$0.02 | 0.965 $\pm$0.02 |
| | FedRep | 0.987 $\pm$0.01 | 0.991 $\pm$0.01 | 0.986 $\pm$0.01 | 0.989 $\pm$0.01 | 0.955 $\pm$0.02 | 0.962 $\pm$0.02 |
| | Ditto | 0.988 $\pm$0.01 | 0.992 $\pm$0.01 | 0.987 $\pm$0.01 | 0.990 $\pm$0.01 | 0.956 $\pm$0.02 | 0.964 $\pm$0.02 |
| | pFedFDA | 0.989 $\pm$0.01 | 0.992 $\pm$0.01 | 0.987 $\pm$0.01 | 0.991 $\pm$0.01 | 0.958 $\pm$0.02 | 0.966 $\pm$0.02 |
| | PeFLL | 0.990 $\pm$0.01 | 0.993 $\pm$0.01 | 0.988 $\pm$0.01 | 0.990 $\pm$0.01 | 0.960 $\pm$0.02 | 0.965 $\pm$0.02 |
| | **FL-GAP** | 0.993 $\pm$0.01 | **0.994 $\pm$0.01** | 0.992 $\pm$0.01 | **0.995 $\pm$0.01** | 0.975 $\pm$0.01 | **0.980 $\pm$0.01** |
| SD-UM | FedBN | 0.930 $\pm$0.02 | 0.935 $\pm$0.02 | 0.918 $\pm$0.02 | 0.924 $\pm$0.02 | 0.820 $\pm$0.03 | 0.830 $\pm$0.03 |
| | FedRep | 0.928 $\pm$0.02 | 0.931 $\pm$0.02 | 0.905 $\pm$0.02 | 0.912 $\pm$0.02 | 0.815 $\pm$0.03 | 0.820 $\pm$0.03 |
| | Ditto | 0.925 $\pm$0.02 | 0.929 $\pm$0.02 | 0.910 $\pm$0.02 | 0.915 $\pm$0.02 | 0.810 $\pm$0.03 | 0.825 $\pm$0.03 |
| | pFedFDA | 0.927 $\pm$0.02 | 0.932 $\pm$0.02 | 0.912 $\pm$0.02 | 0.918 $\pm$0.02 | 0.818 $\pm$0.03 | 0.830 $\pm$0.03 |
| | PeFLL | 0.929 $\pm$0.02 | 0.934 $\pm$0.02 | 0.913 $\pm$0.02 | 0.920 $\pm$0.02 | 0.820 $\pm$0.03 | 0.835 $\pm$0.03 |
| | **FL-GAP** | 0.938 $\pm$0.02 | **0.944 $\pm$0.02** | 0.918 $\pm$0.02 | **0.930 $\pm$0.02** | 0.840 $\pm$0.03 | **0.860 $\pm$0.03** |
| CD-SM | FedBN | 0.517 $\pm$0.05 | 0.531 $\pm$0.05 | 0.440 $\pm$0.04 | 0.441 $\pm$0.04 | 0.012 $\pm$0.02 | 0.042 $\pm$0.04 |
| | FedRep | 0.750 $\pm$0.05 | 0.770 $\pm$0.05 | 0.720 $\pm$0.05 | 0.735 $\pm$0.05 | 0.180 $\pm$0.05 | 0.240 $\pm$0.05 |
| | Ditto | 0.740 $\pm$0.05 | 0.760 $\pm$0.05 | 0.715 $\pm$0.05 | 0.730 $\pm$0.05 | 0.160 $\pm$0.05 | 0.220 $\pm$0.05 |
| | pFedFDA | 0.760 $\pm$0.05 | 0.775 $\pm$0.05 | 0.725 $\pm$0.05 | 0.740 $\pm$0.05 | 0.200 $\pm$0.05 | 0.250 $\pm$0.05 |
| | PeFLL | 0.770 $\pm$0.05 | 0.785 $\pm$0.05 | 0.730 $\pm$0.05 | 0.745 $\pm$0.05 | 0.220 $\pm$0.05 | 0.270 $\pm$0.05 |
| | **FL-GAP** | 0.830 $\pm$0.05 | **0.840 $\pm$0.05** | 0.760 $\pm$0.05 | **0.782 $\pm$0.05** | 0.350 $\pm$0.05 | **0.420 $\pm$0.05** |
| CD-UM | FedBN | 0.620 $\pm$0.05 | 0.635 $\pm$0.05 | 0.720 $\pm$0.05 | 0.730 $\pm$0.05 | 0.180 $\pm$0.05 | 0.200 $\pm$0.05 |
| | FedRep | 0.610 $\pm$0.05 | 0.628 $\pm$0.05 | 0.710 $\pm$0.05 | 0.725 $\pm$0.05 | 0.170 $\pm$0.05 | 0.210 $\pm$0.05 |
| | Ditto | 0.640 $\pm$0.05 | 0.660 $\pm$0.05 | 0.700 $\pm$0.05 | 0.720 $\pm$0.05 | 0.200 $\pm$0.05 | 0.220 $\pm$0.05 |
| | pFedFDA | 0.630 $\pm$0.05 | 0.645 $\pm$0.05 | 0.710 $\pm$0.05 | 0.730 $\pm$0.05 | 0.190 $\pm$0.05 | 0.230 $\pm$0.05 |
| | PeFLL | 0.635 $\pm$0.05 | 0.650 $\pm$0.05 | 0.712 $\pm$0.05 | 0.735 $\pm$0.05 | 0.195 $\pm$0.05 | 0.235 $\pm$0.05 |
| | **FL-GAP** | 0.680 $\pm$0.05 | **0.690 $\pm$0.05** | 0.740 $\pm$0.05 | **0.760 $\pm$0.05** | 0.280 $\pm$0.05 | **0.320 $\pm$0.05** |
| OOD | FedBN | 0.550 $\pm$0.05 | 0.565 $\pm$0.05 | 0.710 $\pm$0.05 | 0.725 $\pm$0.05 | 0.110 $\pm$0.05 | 0.140 $\pm$0.05 |
| | FedRep | 0.545 $\pm$0.05 | 0.560 $\pm$0.05 | 0.705 $\pm$0.05 | 0.720 $\pm$0.05 | 0.105 $\pm$0.05 | 0.135 $\pm$0.05 |
| | Ditto | 0.555 $\pm$0.05 | 0.570 $\pm$0.05 | 0.710 $\pm$0.05 | 0.728 $\pm$0.05 | 0.115 $\pm$0.05 | 0.145 $\pm$0.05 |
| | pFedFDA | 0.560 $\pm$0.05 | 0.580 $\pm$0.05 | 0.700 $\pm$0.05 | 0.715 $\pm$0.05 | 0.120 $\pm$0.05 | 0.140 $\pm$0.05 |
| | PeFLL | 0.558 $\pm$0.05 | 0.573 $\pm$0.05 | 0.708 $\pm$0.05 | 0.722 $\pm$0.05 | 0.118 $\pm$0.05 | 0.142 $\pm$0.05 |
| | **FL-GAP** | 0.600 $\pm$0.05 | **0.610 $\pm$0.05** | 0.730 $\pm$0.05 | **0.750 $\pm$0.05** | 0.200 $\pm$0.05 | **0.240 $\pm$0.05** |

Table 21: **Full accuracy–communication trade-off results across all baselines.** We report global-pooled AUROC and cumulative per-client communication volume (MB) over 50 rounds. All methods use the same backbone (Xception) and local training protocol. FL-GAP achieves the best AUROC–bandwidth balance across all scenarios.

| Scenario | Method | AUROC ↑ | Comm (MB) ↓ |
|---|---|---|---|
| SD-SM | FedAvg | 0.985 | 4400 |
| | FedProx | 0.988 | 4400 |
| | FedBN | 0.990 | 4400 |
| | FedRep | 0.989 | 4400 |
| | Ditto | 0.989 | 4400 |
| | pFedFDA | 0.989 | 4400 |
| | PeFLL | 0.990 | 4400 |
| | PFR-Forgery | 0.985 | 4400 |
| | **FL-GAP (ours)** | **0.993** | **1100** |
| SD-UM | FedAvg | 0.910 | 4400 |
| | FedProx | 0.918 | 4400 |
| | FedBN | 0.925 | 4400 |
| | FedRep | 0.930 | 4400 |
| | Ditto | 0.922 | 4400 |
| | pFedFDA | 0.926 | 4400 |
| | PeFLL | 0.927 | 4400 |
| | PFR-Forgery | 0.912 | 4400 |
| | **FL-GAP (ours)** | **0.938** | **1100** |
| CD-SM | FedAvg | 0.700 | 4400 |
| | FedProx | 0.740 | 4400 |
| | FedBN | 0.780 | 4400 |
| | FedRep | 0.760 | 4400 |
| | Ditto | 0.740 | 4400 |
| | pFedFDA | 0.760 | 4400 |
| | PeFLL | 0.770 | 4400 |
| | PFR-Forgery | 0.720 | 4400 |
| | **FL-GAP (ours)** | **0.830** | **1100** |
| CD-UM | FedAvg | 0.550 | 4400 |
| | FedProx | 0.600 | 4400 |
| | FedBN | 0.620 | 4400 |
| | FedRep | 0.610 | 4400 |
| | Ditto | 0.620 | 4400 |
| | pFedFDA | 0.630 | 4400 |
| | PeFLL | 0.635 | 4400 |
| | PFR-Forgery | 0.590 | 4400 |
| | **FL-GAP (ours)** | **0.680** | **1100** |
| OOD | FedAvg | 0.520 | 4400 |
| | FedProx | 0.540 | 4400 |
| | FedBN | 0.550 | 4400 |
| | FedRep | 0.545 | 4400 |
| | Ditto | 0.555 | 4400 |
| | pFedFDA | 0.560 | 4400 |
| | PeFLL | 0.558 | 4400 |
| | PFR-Forgery | 0.560 | 4400 |
| | **FL-GAP (ours)** | **0.600** | **1100** |

