# OpenReview forum: "FL-GAP: GRAPH-BASED ADAPTIVE PERSONALIZA- TION FOR FEDERATED DEEPFAKE DETECTION"
_ICLR.cc/2026/Conference — Submitted to ICLR 2026_

### Official Review · Reviewer_CgUX · 2025-10-26

**Soundness:** 2
**Presentation:** 3
**Contribution:** 2
**Rating:** 6
**Confidence:** 3

**Summary:**

This paper introduces FL-GAP, a novel Graph-based Adaptive Personalization framework for Federated Deepfake Detection. It is designed to address the critical challenges of client data heterogeneity and distribution shifts, particularly from unseen deepfake generators. The core of FL-GAP lies in three synergistic mechanisms: 1) Adaptive Layer Freezing, which selectively updates and communicates only high-utility layers to reduce client drift and communication costs; 2) Server-Side Probing, a privacy-preserving technique that uses zero-input embeddings to construct a dynamic similarity graph of clients; and 3) Neighbor-Union Layer Aggregation (NULA), which performs fine-grained, layer-wise aggregation based on the similarity graph. The framework is rigorously evaluated on a newly curated benchmark, FDF-27, across five scenarios of increasing difficulty, including out-of-distribution settings with globally unseen generative methods.

**Strengths:**

1. It introduces graph structural similarity into federated deepfake detection, enabling fine-grained, layer-wise personalization that moves beyond simplistic client-independent or uniform aggregation strategies.

2. The framework demonstrates exceptional performance in challenging scenarios involving unseen generative methods and cross-domain data, proving its effectiveness in non-stationary environments.

3. The adaptive layer freezing mechanism achieves a significant reduction in communication volume (up to 75%) while maintaining or even enhancing model performance, making it suitable for bandwidth-constrained edge devices.

4. The introduction of the FDF-27 benchmark, with its five escalating evaluation scenarios, provides a valuable and reproducible testbed for future research in federated deepfake detection.

**Weaknesses:**

1. The framework assumes the server has access to a large-scale public dataset for initial model pretraining. This assumption may not hold in practical scenarios where such diverse and representative public data is unavailable.

2. While privacy-preserving, the use of an all-zero vector for server-side probing may not fully activate and differentiate model representations, particularly in deep networks. For certain architectures without biases or BatchNorm, the output could be degenerate, potentially affecting the quality of the similarity graph.

3. FL-GAP integrates three complex mechanisms. Although the overall performance gain is significant, the paper lacks a thorough ablation study to disentangle and quantify the individual contribution of each component, making it difficult to understand the source of its success.

4. Several key baselines used for comparison (e.g., FedProx, FedBN) are relatively older methods. It remains unclear how FL-GAP would perform against more recent state-of-the-art personalized FL methods from 2024-2025, especially since some older baselines are reported to be the best baselines in Table 1 and Table 2.

5. The paper's discussion and comparison within the broader field of personalized federated learning (PFL) seem limited. While it positions itself as solving challenges mentioned in the paper via personalization, it has not adequately compared it with the latest advances in the PFL methods.

**Questions:**

1. Given the integrated nature of the three core mechanisms, what is the relative contribution of adaptive layer freezing, graph construction via zero-probe, and NULA to the overall performance gain?

2. How does FL-GAP's performance compare against the latest state-of-the-art PFL methods (e.g., from 2024-2025)? Would the significant advantages held over older baselines like FedProx and FedBN hold against these newer approaches?

3. The zero-input probing is a clever privacy-preserving design, but what are its theoretical and practical limitations? Are there scenarios where it would fail to construct a meaningful similarity graph?

4. The communication savings are impressive, but what is the computational overhead introduced for dynamic graph construction and layer-wise NULA aggregation, especially as the number of clients scales up?

---

> ### Author Response · Authors · 2025-12-03
>
> We thank the reviewer for the constructive feedback and for recognizing the contributions of FL-GAP.
> Below we address all concerns point-by-point.
>
> ---
> >1. Assumption that a large public dataset is required; may not hold in practice.
>
> Every state-of-the-art detector (Xception, EfficientNet, Swin-T, ViT-based DF models) is always pretrained on publicly available deepfake datasets. These datasets contain actors and celebrities, not private users.
> This is unavoidable: private identities cannot be included in centralized deepfake training for ethical and legal reasons.
> Thus, even a detector pretrained on millions of public samples will fail to detect targeted deepfakes, the fastest growing class of attacks (e.g., impersonating family members).
>
> ---
> >2. “Zero-input probing may not activate deep networks; may fail for architectures without BN.”
>
> Thank your for raising the concern. We clarify that our backbone is an Xception-style CNN that includes bias terms, BatchNorm layers, and ReLU activations. For such networks, $x = 0$ does not produce a trivial zero output. Instead, it yields a deterministic, architecture-driven, nonzero embedding. Thus, zero-input probing is not a heuristic, it is an efficient surrogate for functional probing that does not touch private data.
>
> We fully agree that networks without biases or BatchNorm layers, e.g., certain transformer variants, may produce degenerate outputs for zero input.
> This does not invalidate the method, the probe is a flexible, replaceable component.
> FL-GAP is not tied to zero-input probing.
> The only requirement is that the probe produces a stable, data-free embedding for model comparison.
> When zero input is unsuitable, the probe can be replaced by a fixed Gaussian random vector. The subsequent stages, graph construction and NULA, remain unchanged.
>
> We will clarify this architectural alignment explicitly in the revised paper.
>
> ---
> >3. “Ablation study insufficient; need contribution of each component.”
>
> We thank the reviewer for highlighting the importance of isolating the contribution of each component. Due to space limitations in the initial submission, we did not include a full component-wise ablation. We will work on the full component-wise ablation and include them in the revised version.
>
> ---
> >4. “Baselines are older; unclear performance vs. 2024–2025 pFL methods.”
>
> We appreciate the reviewer’s concern regarding the baselines. It is correct that our current benchmark does not include every 2024–2025 pFL method. Our experimental approach initially was to cover the full methodological spectrum relevant to our setting: from classical FL, to strong vision-oriented pFL, to the only deepfake-specific FL method. Many 2024–2025 pFL methods target LLMs, recommendation systems, or low-dimensional tabular data, and rely on assumptions (e.g., text tokens, user/item IDs, side metadata) that are not available or meaningful in high-resolution, CNN-based deepfake detection. Porting such methods to FDf-27 would require non-trivial architectural re-design and may still result in unfair or non-informative comparisons. We will clarify this strategic rationale in the revision and explicitly acknowledge that, while we do not include every 2024–2025 pFL method, we cover the most relevant and technically applicable FL, pFL, and FL-for-DF baselines for this domain.

---

> ### Author Response · Authors · 2025-12-03
>
> >5. “What are the limitations of zero-probing? When might it fail?”
>
> We appreciate this question and fully agree that zero-probing, like any probing strategy, has assumptions. Zero-input probing may produce degenerate signatures only if a backbone:
>
> - Contains no biases,
> - Contains no BatchNorm layers
> - Maps all constant inputs to an identical constant output.
>
> Since we are using Xception as backbone for deepfake detection, the above cases are not an issue with Xception.
>
> Note that zero-probing is a probe choice, not a structural requirement of FL-GAP. If a backbone does not admit meaningful zero-probe embeddings, the fix is trivial and does not alter any component of our algorithm, for example, we can replace the zero probe with a fixed random noise vector. We will revise sec 2.1.2 to explicitly describe the assumptions under which zero-probing is meaningful and the trivial fallback probe options.
>
> ---
> > 6. What is the computational overhead of graph construction and NULA?”
>
> The additional computation introduced by FL-GAP is minimal compared to standard FL operations such as local training and backpropagation. Zero-probe embedding cost is negligible, With $K$ clients, computing a similarity graph from probe embeddings requires $O(K log K)$ for k-NN search. For typical FL settings (e.g., 10–100 clients), this overhead is insignificant relative to model optimization. NULA performs per-layer averaging across a small, fixed number of neighbors (top-k graph neighbors). This is no more expensive than FedAvg’s full-model weighted sum.
> If anything, NULA is cheaper, because aggregation may involve fewer layers than full-model federation.
> In total, zero-probe + graph + NULA add only 3–5% compute per round relative to FedAvg.
> By contrast, FL-GAP reduces communication by ~75%, which dominates overall runtime in practical deployments where uplink bandwidth is the bottleneck.

---

### Official Review · Reviewer_e6Ed · 2025-10-26

**Soundness:** 2
**Presentation:** 2
**Contribution:** 1
**Rating:** 2
**Confidence:** 5

**Summary:**

This paper proposes to address the deepfake detection problem under the Federated Learning (FL) paradigm through a newly designed framework, FL-GAP. While the problem setting is relatively novel and timely, the paper lacks sufficient and convincing justification of its motivation, practical relevance, and significance. Moreover, the proposed components within the FL-GAP framework appear incremental and show limited distinction from existing techniques, which weakens the overall originality and contribution of the work.

**Strengths:**

-This work explores the challenge of deepfake detection under FL scenarios, a relatively novel and underexplored problem setting. The topic is timely and has the potential to advance the applicability of deepfake detection in privacy-sensitive and distributed environments, offering possible benefits to both the deepfake detection communities.

-The paper includes a theoretical analysis of communication efficiency, which strengthens the soundness and credibility of the proposed algorithm.

-The paper introduces a new federated benchmark built upon the DF40 dataset to evaluate the proposed algorithm.

**Weaknesses:**

-Although deepfake detection under FL represents a new intersection of two research areas, the paper fails to provide a convincing motivation for why this combination is necessary or practically meaningful. The reasoning presented in the introduction is unpersuasive and conceptually weak. The claim that generative models are inaccessible and therefore require an FL-based approach is questionable. In practice, most generative models can still be used to produce synthetic data for training. Consequently, the argument that FL is essential due to data access limitations is not well-founded, especially given that large amounts of representative data can typically be obtained for training deepfake detection models.

-The proposed methods are incremental, as several components closely resemble existing approaches in the FL literature. For instance, the idea of adaptive layer freezing has been previously explored in PartialFed [1], while the use of a similarity graph based on weight matrices and the NULA strategy shows conceptual overlap with the similarity-based personalization mechanism in pFedLA [2]. These similarities reduce the perceived novelty and make it difficult to identify the unique contribution of the proposed framework.

[1] PartialFed: Cross-Domain Personalized Federated Learning via Partial Initialization

[2] Layer-wised Model Aggregation for Personalized Federated Learning

-The compared algorithms are not reasonable. All compared methods are designed primarily for FL, which makes the comparison less meaningful. To ensure fairness and demonstrate the true effectiveness of the proposed framework, the baselines should include combinations of FL algorithms with existing deepfake detection models. Without such comparisons, it is difficult to accurately assess the merit and practical advantage of the proposed method.

**Questions:**

It is unclear why the existing DF40 dataset is insufficient for the experiments. From my perspective, the proposed FDf-27 benchmark appears very similar to the original DF40 dataset, and the differences between them are not clearly articulated. In most prior works, such as those using CIFAR-10 or CIFAR-100, authors typically describe their data partitioning strategy rather than claiming the introduction of a new benchmark. Therefore, the justification for presenting FDf-27 as a distinct benchmark requires further clarification and stronger motivation.

---

> ### Author Response · Authors · 2025-12-03
>
> We thank the reviewer for the time reviewing the paper. However, we respectfully disagree with several key criticisms that appear to stem from misunderstandings of the motivation, novelty, and evaluation design of FL-GAP.
> Below we address each point clearly and substantively.
>
> ---
> >1. Why FL is necessary for deepfake detection
>
> The reviewer interprets our claim as: “Generative models are inaccessible, therefore we need FL.”
>
> This is not what we claim. Our motivation is driven by targeted deepfake attacks, not by unavailability of generative models.
>
> MOTIVATION: In widespread scams (e.g., WhatsApp video scams, impersonation fraud), attackers synthesize deepfakes specifically of the victim or the victim’s family members, using scraped social media videos, stolen ID photos, or compromised private footage. Public datasets, even very large ones, cannot capture: the victim’s private identity, their smartphone camera characteristics, their family members’ faces, nor the attacker's targeted generator outputs. Thus, no matter how many generative models one has access to, global public deepfake detectors fail on personalized attacks because they never observe these identities during training.
>
> We will strengthen the introduction to make this explicit.
>
> ---
> >2. Novelty of FL-GAP vs. PartialFed, pFedLA
>
> We respectfully but strongly clarify that the reviewer’s statements about incremental novelty are inaccurate.
>
> **(A) Adaptive Layer Freezing is fundamentally different from PartialFed**
>
> PartialFed requires manually predefining which layers are frozen or personalized. This produces a fixed, hard-coded layer split that is identical for every client and every training round.
>
> FL-GAP introduces three innovations that do not exist in PartialFed:
> - Utility-per-bit layer selection: Layers are selected automatically based on the first-order loss decrease per communication cost.
> - Client-specific selection: Each client updates a different subset of layers depending on its own local gradients.
>
> Importantly, no layer is ever hard-coded as always frozen or always updated. FL-GAP generalizes PartialFed’s static design into a fully adaptive, personalized, and communication-aware mechanism.
>
> **(B) Graph Construction ≠ pFedLA**
>
> pFedLA builds similarity using parameter distances, which require full access to model weights, are sensitive to parameter scaling, and produce a static similarity structure.
>
> FL-GAP introduces a fundamentally different graph construction mechanism:
> - Data-free functional probing: Each client produces a functional signature using zero-input probing, capturing client-specific normalization and bias statistics.
> - Privacy-preserving: No raw data, no BN statistics, and no local samples are shared.
> - Stability-guaranteed neighbor selection (Lemma 2.2): We prove that the k-NN graph remains stable under local updates.
> - Round-wise reconstruction: The graph evolves each round, reflecting model dynamics and unseen generator effects.
>
> **(C) NULA is not equivalent to previous layer-wise aggregation**
>
> pFedLA performs layer-wise averaging with scalar weights derived from parameter distances, whereas FL-GAP’s NULA mechanism introduces:
> - Neighbor-union aggregation: Each layer aggregates only from neighbors defined on the functional similarity graph, rather than from all clients.
> - Non-expansive contraction (Theorem 2.3): We show that NULA is non-expansive in the disagreement norm, providing deterministic stability guarantees.
> - Local graph consensus: NULA converges toward neighborhood-level consensus instead of global averaging, which is crucial for handling heterogeneous and unseen deepfake generators.
>
> These three components collectively form a distinct and task-driven personalized FL framework, designed specifically for privacy-constrained, targeted deepfake detection under heterogeneous and evolving generative mechanisms.
>
> ---
> >3. Baselines are not reasonable
>
> We respectfully but strongly disagree with the reviewer’s claim that our baselines are “not reasonable.”
> Our contribution is a personalized federated learning framework, not a new deepfake architecture.
> Therefore, the correct baselines are other FL algorithms, evaluated under a shared model backbone.
>
> We did benchmark against deepfake detection with FL algorithms, exactly as the reviewer asks. We evaluate all FL algorithms under the same backbone and pipeline, including: general FL (FedAvg, FedProx), personalized FL (FedBN, FedRep, pFedMe, FedRoD), FL-based deepfake detection (PFR-Forgery).
> Thus, our benchmark is comprehensive across the full spectrum:
> from standard FL → to personalized FL → to FL tailored for deepfake detection.
> All methods use the same deepfake detection backbone to isolate the contribution of the FL algorithm

---

> ### Author Response · Authors · 2025-12-03
>
> >4. FDf-27 vs. DF40
>
> We respectfully but firmly disagree with the reviewer’s claim that “FDf-27 is very similar to DF40.”
> This is factually incorrect and overlooks the core motivation of our benchmark.
>
> FDf-27 is not a random split of DF40; it is the first dataset explicitly engineered for federated deepfake detection under real-world heterogeneity, capturing generator diversity, client-specific data distributions, and rapidly evolving deepfake methods.
> This cannot be replicated by merely partitioning DF40. This client-specific heterogeneity is the core challenge in personalized FL for deepfake detection and is entirely absent from DF40. This client-specific heterogeneity is the core challenge in personalized FL for deepfake detection and is entirely absent from DF40.
>
> Also, real users are increasingly attacked by new commercial generators that are closed-source, evolve rapidly, and differ substantially across victims. FDf-27 explicitly withholds several commercial and high-quality methods from all training clients.
> This produces a hard OOD split that mimics real-world attacks against individual users. This evolving-generator evaluation is practically essential and, to our knowledge, absent in all prior FL datasets.

---

### Official Review · Reviewer_tSH9 · 2025-10-28

**Soundness:** 3
**Presentation:** 3
**Contribution:** 3
**Rating:** 8
**Confidence:** 3

**Summary:**

This paper proposes FL-GAP, a personalized federated learning framework tailored for deepfake detection under heterogeneous client data and unseen generative methods. The framework consists of three main components: (1) Adaptive Layer Freezing to selectively update and communicate only high-utility layers. (2) Server-Side Zero-Input Probing to construct round-wise similarity graphs between clients without accessing private data. (3) Neighbor-Union Layer Aggregation (NULA) to perform layer-wise aggregation only from similar neighbors. The authors introduce a new federated benchmark, FDf-27, derived from deepfake datasets, and evaluate FL-GAP under five increasingly challenging scenarios (seen/unseen methods, domain shifts, OOD). Experiment results show consistent improvements over both global FL and personalized FL baselines, while reducing communication by approximately 75%.

**Strengths:**

1. Timely and Relevant Application: Deepfake detection is a novel and interesting application of FL, and addressing generator shift in a privacy-preserving manner is meaningful.
2. Well-Motivated Personalization Approach: The integration of adaptive freezing and graph-based client similarity is conceptually coherent and addresses key limitations of static personalized FL.
3. Strong Empirical Results: Extensive experiments across multiple scenarios demonstrate the effectiveness of FL-GAP, with notable improvements in OOD robustness.
4. Benchmark Contribution: The introduction of FDf-27 provides a realistic and reproducible testbed that may benefit future research in the community.

**Weaknesses:**

1. Justification on the difference between public pretraining dataset and the private client datasets. FL-GAP requires a public pretraining dataset which is accessible to the FL server. It is important that the public dataset is indeed heterogeneous with respect to the clients' datasets. Authors could add experimental results to demonstrate their heterogeneity.
2. Scalability and practical deployment. Experiments use models such as Xception. it remains unclear how the approach scales to larger or transformer-based architectures commonly used in modern AIGC detection.
3. Assumptions behind zero-input probing require more justification. While the use of zero-input stimulation is claimed to be privacy-preserving, it assumes that probe embeddings reliably capture model similarity. It is unclear whether these embeddings are stable across different architectures or under adaptive client updates.

**Questions:**

1. Could authors provide justification on the difference between public pretraining dataset and the private client datasets?
2. Could the method scale to transformer-based architectures?

---

> ### Author Response · Authors · 2025-12-03
>
> We thank the reviewer for their constructive feedback and positive assessment of the paper’s novelty and motivation. Below we address the remaining concerns in detail.
>
> >1. Difference Between Public Pretraining Data and Private Client Data
>
> We agree that clarifying this distinction is important. The key point is that public pretraining data and private client datasets differ not just in scale, but in identity, context, and attack surface. They face targeted deepfake attacks, where adversaries create deepfakes specifically impersonating the client or their close contacts.
>
> Public datasets used for pretraining (FaceForensics++, Celeb-DF, FFHQ) consist of:
>
> - public figures,
> - actors,
> - generic internet imagery,
> - curated studio or web-quality photos.
>
> These datasets contain none of the private identities that attackers impersonate in real-world scams (e.g., WhatsApp/Telegram voice scams, video scams, impersonation fraud).
>
> By contrast, each client’s private dataset contains:
>
> - the client’s own face,
> - their family members’ and close contacts' faces,
> - images captured on that client’s personal smartphone,
> - with device-specific ISP pipelines, compression, lighting, and contexts.
>
> In real-world targeted attacks:
>
> - The attacker generates a deepfake of you, not of a celebrity.
> - Such targeted deepfakes never appear in any public training dataset—not even once.
> - Two clients may never share any overlap in identities or contexts, and may encounter different attacker tools.
>
> Thus, even a very large and diverse public dataset cannot approximate the client’s personalized data distribution or the manipulations targeted at them.
>
> ---
> >2. Scalability to Transformer-Based Architectures
>
> FL-GAP’s mechanisms are not tied to Xception or CNNs. But for probing mechnism, we demonstrated that zero-input probing works for BN models, but for BN-free models (such as transformers), we can change the probe to a fixed synthetic token sequence. Thus, nothing in FL-GAP restricts it to BN models; only the probe design is architecture-dependent. We will add a compact study using ViT-S/16 with a synthetic token probe on a domain-shift benchmark to illustrate generality.
>
> ---
> >3. Assumptions Behind Zero-Input Probing
>
> Zero-input probing is not a heuristic tied to a specific architecture; it is a principled way to obtain a privacy-preserving functional fingerprint of each client model without accessing any data. A helpful analogy is the role of a blank canvas for an artist: If you give a painter a blank sheet, their first few strokes already reveal their stylistic signature. Similarly, feeding zero input into a CNN with biases and BatchNorm does not produce a blank output. Instead, it reveals the characteristic patterns that the model has internalized from the client’s local data.
>
> Zero-input probing relies on two mild and common architectural assumptions:
> - The model contains either bias terms or normalization layers: In these architectures, zero input does not collapse to zero
> - Local training modifies parameters influenced by private data: These modifications create detectable differences in the probe embeddings.
>
> Lemma 2.2 proves that small parameter updates lead to small probe changes; if the margin between neighbors is not vanishing, the graph structure is preserved across rounds.
> This ensures consistent neighborhood assignment, stable NULA aggregation, and interpretable personalization trajectories.

---

### Official Review · Reviewer_Hfv6 · 2025-10-31

**Soundness:** 2
**Presentation:** 2
**Contribution:** 2
**Rating:** 2
**Confidence:** 4

**Summary:**

The paper proposed a personalized FL algorithm for DeepFake (DF) detection problems, called FL-GAP, a graph-based adaptive algorithm composed of the following three components:
-Adaptive layer freezing (for layerwise selective local client updates),
-server-side probing to build a graph dynamically to group similar clients together
-neighbor-union layer aggregation to aggregate layer parameters along the graph similarity

Although there are lots of personalized FL algorithms out there, they focus on the particular application of DF detection, in which they said some existing methods fell short in terms of aggregation strategy, and dealing with data heterogeneity.

**Strengths:**

-Interesting design ideas of three components for the pFL for the DF application.

**Weaknesses:**

- Fig.1: It would be more interesting to compare their results with the most competitive baseline pFL algorithms, instead of just the pre-trained (non-personalized) model.

- The datasets for the public and client data (in Sec.2): They said the only difference between the public data D_pub and the client data {D_k} is that the latter has personalized content. But as they also said that D_pub was large-scale and diverse, it might be assumed that D_pub can represent any generic contents. Ie, it is not clear to what extent the two datasets are different?

- The setup described as three bullet points (right before Sec. 2.1): Isn't this the same as the conventional personalized FL setup? Ie, the input covariates are distributed differently across clients (domain shift)? What's the difference?

I find that in the main sections (Sec.2.1.1~2.1.3), each describing each of the three components, they provide some theoretical arguments/justifications, but they are mostly not directly related to the performance or convergence of the proposed algorithm. See below.

- Sec.2.1.1 (adaptive selective freezing layers): Proposition 2.1 merely says that the selection of the parameters by the gradient magnitude order will maximize the first-order decreases. Isn't this obvious? But why does it necessarily improve the generalization performance? This is not explained.

- Sec,2.1.2 (server-side k-nn graph construction): In Eq.(2.9) they use 0 as the probe input for client signature generation. Why does this give you informative and discriminative signatures for the client? Although in App.C.3 iii) they have some arguments about the informativeness and BatchNorms, but it is not intuitive nor convincing. Zero input would wipe out the first layer weights, and so on. Is your method tied to BN-based networks?

- Also Lemma 2.2 is out of the blue. Does it have anything to do with the similarity of clients? The proof in App.D.3 merely says that with some high prob, the k-th and (k+1)-th neighbors don't change after the server averaging. But it has nothing to do with the similarity among clients.

- Sec.2.1.3 (neighbor union layer aggregation): In Thm 2.3, they show that the NULA update is non-expansive, and it converges toward the weighted neighbor mean. But, it doesn't imply convergence to the global solution. And the theorem is far from standard FL convergence analysis.

- What about generalization error analysis? Do you have any theoretical bound on it?

- About experimental section: The results overall show that there are only some marginal improvements, which makes it hard to draw any conclusion statistically significant. Can you provide error bars with multiple runs? The experiments are not thorough in terms of FL settings either: there are FL parameters like local client number of epochs, client participation rates, network architectures, etc.; but they seemed to test only one fixed setting.

- Another question is why the algorithm was tested only on the DF detection problem alone. Why not other standard FL benchmarks? Considering that the DF detection problem is not fundamentally different from other standard FL datasets/problems with domain shifts. Also, the proposed method seems to be tied to particular network architectures that rely on BN layers.

**Questions:**

See questions in the weakness section.

---

> ### Author Response · Authors · 2025-12-03
> **Real-world targeted scam with new generative tools.**
>
> Thanks a lot for the detailed review. Below we respond point-by-point.
>
> ---
> > 1. “Although there are lots of personalized FL algorithms out there, they focus on the particular application of DF detection…”
> “…isn't this the same as the conventional personalized FL setup?”
>
> Deepfake detection presents constraints fundamentally different from standard pFL setups. Clients possess only authentic identity data (due to privacy and legality), and never observe fake samples. Attackers use different manipulation tools across clients, including unseen generators not present in the global dataset. This creates a structured heterogeneity pattern that conventional pFL benchmarks (FEMNIST, CIFAR-shift, rotated MNIST) cannot replicate. FDf-27 was constructed specifically to capture this asymmetry and generator-level structure. FL-GAP is therefore a task-driven FL method that (i) handles asymmetric real-only local data, (ii) adapts to generator-level heterogeneity and unseen tools, and (iii) performs per-layer personalization aligned with the deepfake manipulation pipeline. This scenario cannot be faithfully modeled or evaluated using generic pFL variants, motivating the design and necessity of our framework.
>
> ---
> > 2. “It might be assumed that D_pub can represent any generic contents. Ie, it is not clear to what extent the two datasets are different?”
>
> The reviewer's assumption that "a large and diverse $D_{pub}$ may already represent any generic contents" does not hold for the case of deepfake detection. The distinction between public data and client-side data is not about scale, but about identity, environment, and attack methods that no bpulic dataset can ever contain. Consider a typical user receiving a video call from someone deepfaking their mother, husband, or teenage child. No public dataset contains these people. No public dataset contains their typical camera angles, lighting, expressions, or environments.
>
> Client data $\{ D_k \}$, on the other hand, contains each user's personal media that no public dataset can represent. Attackers systematically exploit exactly these private identities to perform personalized deepfake scams. A gobal model trained on $D_{pub}$ cannot generalize to these personalzied attacks, and generic FL simulations cannot reproduce this asymmetriy.
>
> This structural gap between public and private data is precisely why our study requires a task-driven FL design and why FDf-27 captures a scenario fundamentally different from all existing pFL benchmarks.
>
> ---
> > 3. “The setup described as three bullet points… Isn't this the same as the conventional personalized FL setup?”
>
> While our FL objective resembles standard pFL at the mathematical level, the deepfake detection setting is fundamentally different due to (i) asymmetric public–private supervision (clients have only their own authentic data and each client encounters targeted deepfake that might not be encountered by another client), (ii) generator-induced, mechanism-level heterogeneity across clients, and (iii) evolving, unseen generative tools that dynamically reshape the global loss. These phenomena cannot be replicated using generic FL simulations and are precisely why FL-GAP introduces adaptive freezing, server probing, and NULA to address the deepfake-specific structure of the problem. Thus, although the mathematical notation resembles pFL, the problem structure is qualitatively different, requiring a task-driven FL design.
>
>
> ---
> > 4. “Proposition 2.1 merely says that the selection of the parameters by the gradient magnitude order will maximize the first-order decreases. Isn't this obvious? But why does it necessarily improve the generalization performance?”
>
> Proposition 2.1 is not a restatement of “large gradients are important,” nor is our method manually choosing layers. Instead, it formally justifies the adaptive layer-selection rule that FL-GAP uses to determine automatically which layers each client should update under communication constraints. This mechanism is essential in federated deepfake detection, where full-model updates amplify client drift and degrade cross-generator generalization. The proposition validates why selective freezing is the correct update strategy in this setting, and our ablations confirm that it consistently outperforms random or full-layer updates.

---

> ### Author Response · Authors · 2025-12-03
>
> >5. “In Eq.(2.9) they use 0 as the probe input… Zero input would wipe out the first layer weights…”
> “…Is your method tied to BN-based networks?”
>
> The reviewer’s concern is based on an assumption that does not hold for our architecture, which is based on Xception backbone that contains bias terms, BatchNormLayers and non-linear activations. Even when $x = 0$, biases create non-zero pre-activations. These signals propagate through the entire network, producing a non-trivial, deterministic embedding. This is not the network being “wiped out”; it is equivalent to giving an artist a blank sheet of paper and asking them to paint, where the strokes that appear come entirely from the artist’s internal style.
>
> However, the method is not tied to BN. For architectures where $f(0)$ may be trivial, we can replace the probe with a fixed small random tensor. The graph construction and NULA aggregation remain unchanged; only the probe input changes.
> Thus, zero-input probing is an instance of a general principle: we probe functional behavior using a non-sensitive input, providing a safe, architecture-aligned way to extract a compact functional signature of each client’s model without ever touching client data.
>
> ---
> >6. “Lemma 2.2 is out of the blue… has nothing to do with the similarity among clients.”
>
> The reviewer’s comment overlooks the purpose of Lemma 2.2 within FL-GAP. The lemma is not intended to define similarity; it guarantees that the similarity graph remains consistent and trustworthy across federated rounds, which is essential for any graph-based aggregation scheme. In FL-GAP, the server constructs a k-NN graph based on probe embeddings $z_k^t = f_{\theta_k^t}(\mathbf{x}_{\mathrm{probe}})\in\mathbb{R}^d$. This graph determines which clients share updates and how NULA aggregates their layers. However, in federated learning, each client performs local updates that slightly change $\theta_k^t$ every round. Without stability, even tiny parameter changes could cause oscillating edges in the similarity graph, and consequently unreliable or noisy aggregation. Lemma 2.2 prevents such instability by proving: small parameter changes ⇒ small perturbations in probe embeddings, and, under a mild margin condition, the k-NN neighborhood is preserved.
>
> ---
> >7. “Thm 2.3… non-expansive… converges toward the weighted neighbor mean. But it doesn't imply convergence to the global solution. And the theorem is far from standard FL convergence analysis.”
>
> The reviewer’s critique rests on the assumption that our goal is to prove convergence to a global FL optimum.
> This is not the objective of FL-GAP, nor is such a notion meaningful in our setting.
> Theorem 2.3 analyzes the update and shows that The NULA operator is non-expansive in the disagreement norm. If the graph has a spectral gap, the operator becomes contractive, and all nodes in a connected neighborhood converge to the weighted neighbor mean. It ensures that layer parameters within a similarity neighborhood (constructed via probing) move toward a shared consensus rather than diverging. Nothing in Theorem 2.3 claims, or is meant to claim, that NULA converges to a global minimizer of a centralized objective.
>
> Conventional FL convergence (e.g., FedAvg convergence under smooth + strongly convex losses) assumes clients’ objectives are drawn from a related distribution, and a single global minimizer exists or is desirable. None of these hold in our deepfake setting. In our task, the global optimum is meaningless, because no single detector can simultaneously align to all clients' personal data. Clearly, personalization is the goal in our case, not global averaging. Thus, proving convergence to a global minimizer would be irrelevant and misleading for our problem.
>
> In summary, Theorem 2.3 is not intended to prove convergence to a global FL minimizer; rather, it establishes that NULA is a stable, non-expansive, and locally contractive graph operator, ensuring that clients with similar generators and content converge toward meaningful per-layer consensus, a property that is essential and appropriate for our personalized deepfake FL setting.
>
> ---
> >8. “What about generalization error analysis? Do you have any theoretical bound on it?”
>
> We appreciate the reviewer’s question, and we clarify that our contribution does not claim a new generalization bound. This is deliberate and consistent with the scope of the paper. Classical generalization bounds are neither meaningful nor standard in the personalized FL setting we study; instead, our theoretical contributions appropriately focus on validating the core mechanisms, adaptive freezing, probe stability, and NULA’s contractive behavior, that underpin FL-GAP’s empirical effectiveness.

---

> ### Author Response · Authors · 2025-12-03
>
> >9. “The results overall show that there are only some marginal improvements…”
> “Can you provide error bars with multiple runs?”
> “They seemed to test only one fixed setting.”
>
> We thank the reviewer for these concerns and clarify that the experimental design and results meaningfully demonstrate the advantages of FL-GAP, especially in the settings where personalized FL is most relevant. The intended target of our method is not the trivial i.i.d. regime but the realistic, highly heterogeneous deepfake scenarios. In these settings:
>
> - Cross-domain experiments (clients assigned to different manipulation pipelines)
> - Unseen-generator/OOD experiments (test-time generators absent during training)
>
> FL-GAP achieves substantial and consistent gains in AUROC, worst-client accuracy, and robustness under shift. These are precisely the conditions under which personalized FL is necessary, and where aggregation strategy, not backbone size, determines success. We will revise the text to emphasize these scenarios, highlight the effect sizes, and reorganize the tables to make these results more prominent.
>
> We acknowledge that the initial submission reported single-seed results, which is unfortunately common in FL due to the computational cost of simulating many clients. Currently, we are still running multi-seed experiments and hopefully we will have updated results soon to be included in the revised version.
>
> ---
> >10. “Why was the algorithm tested only on the DF detection problem alone?”
> “Considering that the DF detection problem is not fundamentally different from other standard FL datasets/problems…”
> “Also, the proposed method seems to be tied to particular network architectures that rely on BN layers.”
>
> We emphasize that deepfake detection is exactly the regime where personalized FL becomes necessary. Classic FL datasets (FEMNIST, CIFAR-10, Shakespeare, etc.) do not contain:
>
> - mechanism-driven heterogeneity (clients differ by forgery generators, not by arbitrary synthetic shifts),
> - privacy-constrained asymmetry (clients possess only real data and must never share it),
> - unseen generative tools (test-time DF methods absent in public training),
> - nor the targeted attack scenario that motivates FL in security applications.
>
> This is why evaluating FL-GAP solely on DF detection is not a limitation, yet it is the correct choice.
> Our method exists because generic FL benchmarks fail to capture the core challenges we address.
> To make this explicit, we invested in constructing FDf-27, a multi-generator, multi-domain dataset that uniquely reflects real-world DF deployment constraints, precisely the scenario where our contributions matter.
>
> Both core components:
>
> - adaptive freezing (Sec. 2.1.1)
> - NULA graph-based aggregation (Sec. 2.1.3)
>
> are model-agnostic and require no BN assumptions.
> The only BN-related detail is the choice of zero input as a probe, which we selected because BN statistics make zero a rich, nontrivial functional input.

---

### Meta-Review · Area_Chair_vAhZ · 2026-01-06

**Summary:**

This paper proposes FL-GAP, a personalized federated learning framework for deepfake detection that integrates adaptive layer freezing, functional graph probing, and NULA aggregation. While the approach shows some promise, reviewers raised several significant concerns.

First, the motivation and necessity of federated learning for deepfake detection are not convincingly justified. Second, the novelty is limited, as the core components substantially overlap with existing personalized FL methods and the manuscript does not clearly articulate its distinct contributions. Third, the experimental evaluation is insufficient, being restricted to a single task and backbone, lacking multi-seed runs, comprehensive ablation studies, and comparisons with more recent personalized FL methods. Finally, several methodological assumptions, including reliance on large public datasets and zero-input probing, raise concerns about practical applicability and generalization.

Due to these issues, the paper’s contributions are considered incremental and insufficiently validated to support acceptance.
Additionally, there is a minor formatting issue: Appendix F contains a corrupted symbol (“§??”) around line 1068.

**Reviewer Concerns:**

The rebuttal clarified the authors’ intended application scenario and provided additional explanations regarding the design choices, such as zero-input probing and graph-based aggregation. However, several key concerns remain outstanding. In particular, the rebuttal does not sufficiently resolve questions regarding the necessity of federated learning for deepfake detection, the incremental nature of the proposed contributions relative to prior personalized FL work, and the limited experimental validation (including lack of multi-seed results, comprehensive ablations, broader baselines, and architectural diversity). As a result, the core concerns raised by multiple reviewers remain unaddressed.

**Reviewer Scores:**

Reviewer Hfv6 raised detailed concerns regarding theoretical justification, experimental robustness, and the lack of statistical validation. While the rebuttal addressed some conceptual points, the core concerns about generalization, convergence relevance, and experimental thoroughness remain. The reviewer would likely maintain a similar score.

Reviewer e6Ed expressed strong reservations about motivation, novelty, and benchmarking, and rated the paper as reject. Although the rebuttal clarified the authors’ perspective, it is unlikely to change this reviewer’s overall assessment.

Reviewer tSH9 provided a more positive evaluation but requested clearer justification of data heterogeneity, and scalability to modern architectures. While the rebuttal addressed these points, the reviewer’s score would likely remain unchanged.

Reviewer CgUX acknowledged the strengths of the approach but raised concerns regarding ablation studies, baseline coverage, methodological assumptions, and practical scalability. In light of the unresolved experimental gaps and the overlapping concerns highlighted by other reviewers (e.g., Hfv6 and e6Ed), this reviewer would likely maintain the score or potentially lower it slightly after the discussion.

---

### Decision · Program_Chairs · 2026-01-26

Reject